# Topologically guided tuning of Zr-MOF pore structures for highly selective separation of C6 alkane isomers

Hao Wang[1], Xinglong Dong [2], Junzhong Lin[3], Simon J. Teat [4], Stephanie Jensen[5], Jeremy Cure[6], Eugeny V. Alexandrov [7], Qibin Xia[1,8], Kui Tan[6], Qining Wang[1], David H. Olson[1], Davide M. Proserpio [7,9], Yves J. Chabal[6], Timo Thonhauser[5,10], Junliang Sun[3], Yu Han [2] & Jing Li[1]

As an alternative technology to energy intensive distillations, adsorptive separation by porous solids offers lower energy cost and higher efficiency. Herein we report a topology-directed design and synthesis of a series of Zr-based metal-organic frameworks with optimized pore structure for efficient separation of C6 alkane isomers, a critical step in the petroleum refining process to produce gasoline with high octane rating. $Zr_6O_4(OH)_4(bptc)_3$ adsorbs a large amount of $n$-hexane but excluding branched isomers. The $n$-hexane uptake is ~70% higher than that of a benchmark adsorbent, zeolite-5A. A derivative structure, $Zr_6O_4(OH)_8(H_2O)_4(abtc)_2$, is capable of discriminating all three C6 isomers and yielding a high separation factor for 3-methylpentane over 2,3-dimethylbutane. This property is critical for producing gasoline with further improved quality. Multicomponent breakthrough experiments provide a quantitative measure of the capability of these materials for separation of C6 alkane isomers. A detailed structural analysis reveals the unique topology, connectivity and relationship of these compounds.

[1] Department of Chemistry and Chemical Biology, Rutgers University, 610 Taylor Road, Piscataway, NJ 08854, USA. [2] Advanced Membranes and Porous Materials Center, Physical Sciences and Engineering Division, King Abdullah University of Science and Technology, Thuwal 23955-6900, Saudi Arabia. [3] College of Chemistry and Molecular Engineering, Peking University, Beijing 100871, China. [4] Advanced Light Source, Lawrence Berkeley National Laboratory, 1 Cyclotron Road, Berkeley, CA 94720, USA. [5] Department of Physics, Wake Forest University, Winston-Salem, NC 27109, USA. [6] Department of Materials Science & Engineering, University of Texas at Dallas, Richardson, TX 75080, USA. [7] Samara Center for Theoretical Materials Science (SCTMS), Samara University, Samara 443011, Russia. [8] School of Chemistry and Chemical Engineering, South China University of Technology, Guangzhou 510641, China. [9] Dipartimento di Chimica, Università degli Studi di Milano, Milano 20133, Italy. [10] Department of Chemistry, Massachusetts Institute of Technology, Cambridge, MA 02139, USA. Correspondence and requests for materials should be addressed to J.L. (email: jingli@rutgers.edu)

Chemical separation accounts for ~50% of the industrial energy use in the United States and 10–15% of the nation's total energy consumption[1]. The separation of hydrocarbons is a crucial process in petrochemical industry for the manufacture of high quality gasoline, plastics, and polymers. For example, the separation of C6 alkane isomers is necessary to produce premium grade gasoline yet difficult because of their similar chemical and physical properties[2]. During the oil refining process, hexane isomers are generated from catalytic isomerization reactions and are subject to separation based on their research octane number (RON). N-hexane (nHEX, RON = 30) needs to be excluded from its branched isomers (RON = 75 or higher) to produce gasoline of high quality. Aside from lowering the energy cost, adsorptive separation through porous media significantly reduces carbon dioxide emission compared to the large scale industrial distillation processes currently used for the separation of C6 alkane isomers. Zeolite 5A (LTA), as the benchmark material for this separation process, is able to adsorb linear alkanes while excluding mono- and di-branched isomers owing to its suitable pore aperture[3]. Adsorptive separation of hexane isomers by zeolite 5A has been employed in industry as a supplement to distillation[4]. However, the relatively low uptake capacity for nHEX (e.g. ~8 wt% at 150 °C and 105 torr) limits its separation efficiency. In addition, the incapability of adsorbing any branched C6 isomers prevents its use for achieving further improved RON by differentiating mono- and di-branched isomers. These limitations of zeolite 5A have motivated researchers to continue to seek new types of porous materials with enhanced performance for this process[5].

Adsorption-based separation of hydrocarbons by porous solids can be divided into two categories according to the separation mechanism: kinetically controlled and thermodynamically controlled process[6,7]. The former is based on the difference in diffusion rate, or in an ideal scenario, on selective molecular exclusion (or sieving), which usually results in high selectivity, as illustrated by two well-known examples: zeolite 5A for the separation of linear and branched alkane isomers and chabazite zeolite (CHA) for the separation of propane and propylene[8]. In contrast, thermodynamically controlled separation is governed by the difference in affinity between distinct, freely diffused adsorbates and the framework. While it is usually less selective than kinetically controlled separation, thermodynamically controlled processes can be advantageous when adsorbates are very similar in size which would be difficult to discriminate by kinetic separation[9].

Metal-organic frameworks (MOFs) have been extensively investigated for gas storage and separation not only because of their high porosity but more importantly, they also offer fascinating tunability with respect to their pore size, shape, and surface functionality[10–14]. These features make them attractive candidates for energy-efficient separation of hydrocarbons via different mechanisms not easily achievable by traditional porous solids[7,15,16]. Research on hydrocarbon separation using MOFs is less mature compared to carbon dioxide capture[17–20] or hydrogen/natural gas storage[21,22], but exciting progress has been made over the past several years[7,23–26]. Unprecedented performance of MOFs has been achieved for industrially important separation of ethylene and acetylene,[24] and propane and propylene[23] mixtures. In the case of separation of C6 alkane isomers,[27] Long and co-workers have demonstrated that $Fe_2(BDP)_3$, a microporous MOF with triangular channels, is able to separate hexane isomers despite its propensity to adsorb all isomers, as shown in their breakthrough data. This is attributed to their differences in van der Waals interaction with the MOF channels. While MOFs have shown strong potential for the separation of alkane isomers[5,28–33], search for new adsorbents with higher adsorption capacity and selectivity that outperform zeolite 5 A is much needed.

In this work we focus on a specific MOF family, namely structures built on zirconium and tetratopic carboxylate linkers, for the following reasons: Zr-based metal-organic frameworks (Zr-MOFs) are a subgroup of MOFs that generally possess high chemical, thermal and water/moisture stability as a result of strong Zr–O bonds and robust multinuclear secondary building units (SBUs, usually $Zr_6$ clusters)[34]. In addition, the structure types and framework stability of Zr-MOFs built on tetratopic linkers largely depend on the geometry of the linkers, which may be rationally designed by judicious selection of organic ligands[35]. For example, a specific **ftw** type structure[36] can form with Zr and a rigid and planar tetratopic linker, featuring three-dimensional (3D) porous frameworks with large cubic cages but small window aperture, which is particularly desirable for molecular separation through selective size sieving. However, all **ftw** type Zr-MOFs reported to date are built on relatively large organic linkers[37–40] (e.g. porphyrin or pyrene based molecules) which result in pore apertures that are too large for a separation process based on molecular exclusion. Since the pore aperture of a **ftw** Zr-MOF is dominated by the distance between adjacent carboxylates of the organic ligand (or adjacent $Zr_6$ SBUs in the MOF structure), we have deliberately selected isophthalate based tetratopic linkers with appropriate molecular dimensions to reduce the distance between adjacent SBUs and consequently the pore aperture. To investigate the intricate relationship between the dimension of a ligand and the resulting MOF topology, we focus our effort on a series of three organic linkers with similar geometry but different aspect ratios[35], 3,3′,5,5′-biphenyltetracarboxylate (bptc), 3,3′,5,5′-azobenzene-tetracarboxylate (abtc), and 2′,5′-dimethyl-[1,1′:4′,1″-terphenyl]-3,3″,5,5″-tetracarboxylate (tptc-(Me)$_2$). We have succeeded in obtaining crystals of all three Zr-MOF compounds after a systematic optimization of synthetic conditions. Two of these structures, Zr-bptc and Zr-abtc, are highly stable frameworks with optimal pore structure for the separation of C6 alkane isomers. Their performance is comparable and, in some aspects, outperforms the benchmark material zeolite 5A.

## Results

**Synthesis and crystal structure.** All three crystalline materials, $Zr_6(\mu_3\text{-}O)_4(\mu_3\text{-}OH)_4(bptc)_3$ (Zr-bptc or compound **1**), $Zr_6(\mu_3\text{-}O)_4(\mu_3\text{-}OH)_4(abtc)_2(OH)_4(H_2O)_4$ (Zr-abtc or compound **2**), and $Zr_6(\mu_3\text{-}O)_4(\mu_3\text{-}OH)_4(tptc\text{-}(Me)_2)(HCOO)_4(OH)_4(H_2O)_4$ (Zr-tptc-(Me)$_2$ or compound **3**) were synthesized by solvothermal reactions (see Methods for details). Compounds **2** and **3** were structurally characterized by single crystal X-ray diffraction (SCXRD) analysis while the crystal structure of compound **1** was determined via powder refinement (Fig. 1 and Supplementary Tables 1–3).

Compound **1** crystallizes in cubic crystal system (space group $Im3$) and features 4,12-c **ftw** topology[37,38]. The structure displays 12-connected $Zr_6(\mu_3\text{-}O)_4(\mu_3\text{-}OH)_4(COO)_{12}$ SBUs, linked together through 4-connected bptc$^{4-}$ ligands to form a 3D framework (Supplementary Figs. 1 and 2). The SBU is composed of six Zr atoms assembled into an octahedron where the eight facets are occupied by $\mu_3\text{-}O^{2-}$ or $\mu_3\text{-}OH^-$ anions. Each Zr atom is coordinated to eight O atoms, four of which belong to four different bptc$^{4-}$ ligands and the remaining four are from $\mu_3\text{-}O^{2-}/OH^-$ groups. Each bptc$^{4-}$ linker is connected to four different SBUs with each carboxylate coordinated by two adjacent Zr atoms in the same SBU in a bi-monodentate fashion. Similar to other reported Zr-MOFs with **ftw** topology, compound **1** contains cubic cage-like pores with $Zr_6$ clusters on the vertices and planar bptc$^{4-}$ linkers on the faces. The cages have a dimension of ~12 Å

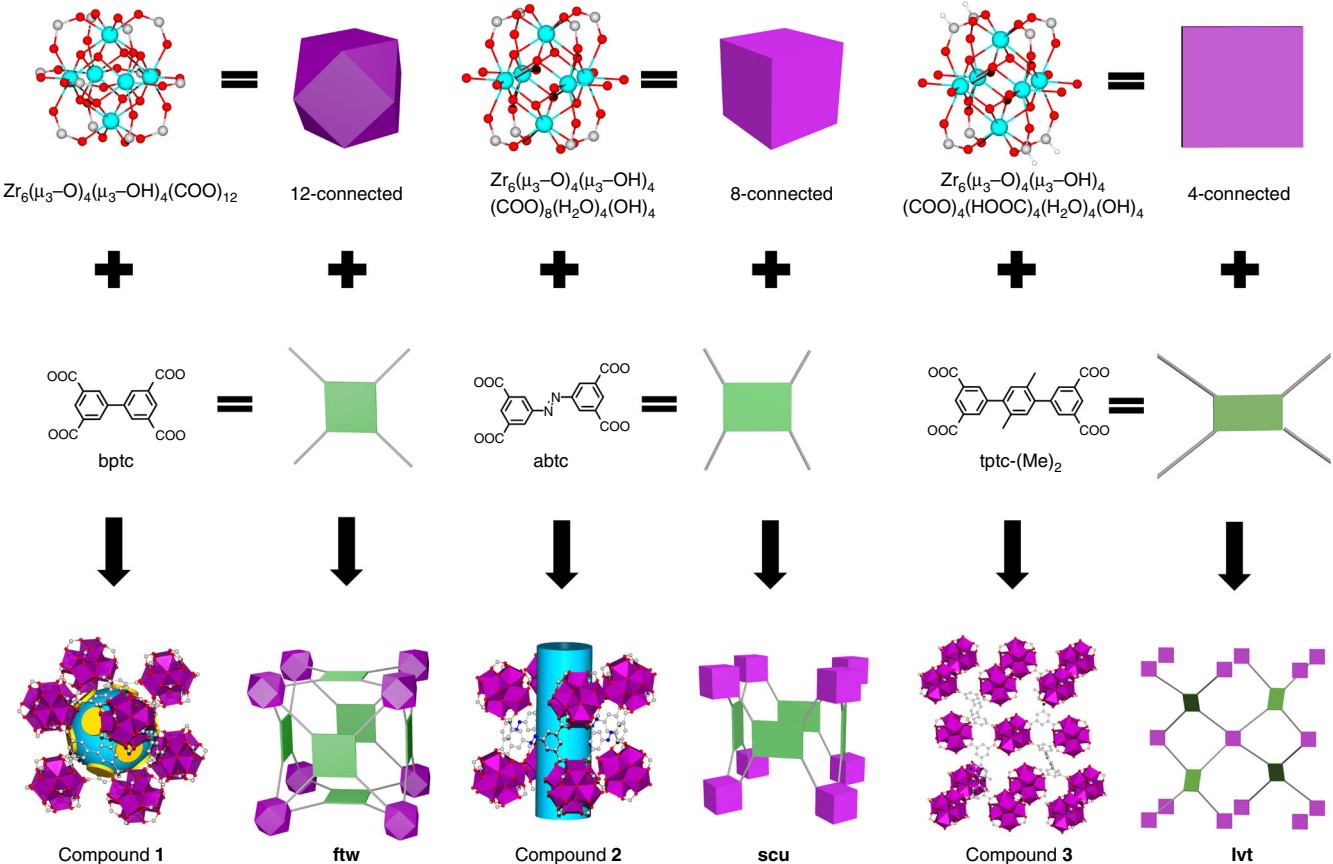

**Fig. 1** Structure analysis of compounds **1–3**. Compounds **1**, **2**, and **3** are built on 12-, 8-, and 4-connected $Zr_6$ clusters, linked by tetratopic organic ligands bptc, abtc, and tptc-(Me)$_2$, forming **ftw**, **scu**, and **lvt** type structures, respectively

and are interconnected through smaller tetrahedral cages located at the twelve edges of the cubic cages. These tetrahedral cages have a window size of ~4.5 Å. It is worth to note that bptc$^{4-}$ is the smallest member of all tetratopic linkers for Zr-MOFs reported to this date (Supplementary Figs. 3 and 4). Compound **1** also represents the first Zr-MOF built on isophthalate-based polytopic ligands[34]. While isophthalate-based organic linkers have been widely used in constructing MOFs with various metals[41], no zirconium MOFs made of such ligands have been reported before the current work. This is likely because of the short distance between the two carboxylates on the same isophthalate moiety which gives rise to an added difficulty in forming extended structures with large $Zr_6$ SBUs and our synthesis suggests a large amount of acid modulator is necessary to obtain crystalline products.

Interestingly, replacing bptc$^{4-}$ by abtc$^{4-}$ did not give rise to an isoreticular compound (Supplementary Fig. 5). Single crystal X-ray diffraction analysis reveals that compound **2** crystallizes in monoclinic crystal system (space group $C2/m$). The structure consists of 8-connected $Zr_6(\mu_3-O)_4(\mu_3-OH)_4(COO)_8$ SBUs that are propagated by 4-connected abtc$^{4-}$ linkers along three dimensions, forming a framework with a rare 4,8-c **scu** topology[31,40,42,43]. The connectivity of the SBU is reduced to 8 from 12 in compound **1** as a result of four out of twelve carboxylate groups being replaced by terminal $H_2O/OH^-$. In addition to the coordination to four O atoms from capped $\mu_3-O^{2-}/OH^-$, each Zr atom at the equatorial position also coordinates to two carboxylate O atoms from two different abtc$^{4-}$ ligands, and another two O atoms from terminal $H_2O/OH^-$ groups. The four remaining coordination sites of each of the two Zr atoms at

the apical position are all taken by carboxylate O atoms from four different abtc$^{4-}$ ligands. The rectangular ligand abtc$^{4-}$ is present in both *trans* and *gauche* conformation. Comparing to the overall connectivity of compound **1**, the abtc$^{4-}$ ligands along the crystallographic *a*-axis are missing in compound **2**, leading to the transformation from cage-like pore to 1D channel with a diameter of ~7 Å. The preference of **scu** topology over **ftw** for compound **2** is due to the increase in aspect ratio of the organic ligands (from 1.45 for bptc$^{4-}$ to 1.78 for abtc$^{4-}$) which results from a change of a nearly square shaped tetratopic ligand to a rectangle shaped ligand.

Further increase in the aspect ratio of the organic ligand tptc-(Me)$_2$ (2.28, where two isophthalate groups are separated apart by a phenyl ring) led to a totally different connectivity in compound **3** (Supplementary Fig. 6). Single crystal X-ray diffraction analysis shows that compound **3** crystallizes in orthorhombic crystal system (space group *Imma*). In this structure, the formula of the SBU changes to $Zr_6(\mu_3-O)_4(\mu_3-OH)_4(COO)_4$, where the $Zr_6$ octahedral core remains but its connectivity further reduces to 4 as a result of replacing four carboxylates in compound **2** by four terminal formate groups. Each of the four equatorial Zr atoms coordinates to eight O atoms where four are from bridging $\mu_3-O^{2-}/OH^-$, two from terminal $H_2O/OH^-$, one from terminal formate and one from a carboxylate of a tptc-(Me)$_2^{4-}$ linker. The two apical Zr atoms coordinate to four bridging $\mu_3-O^{2-}/OH^-$, two formate groups and two carboxylates from two distinct tptc-(Me)$_2^{4-}$. The terminal formate groups adopt a bi-monodentate coordination mode, similar to carboxylates from tptc-(Me)$_2^{4-}$ linkers. The connection mode of the 4-connected SBU in compound **3** resembles that of the paddle-wheel dinuclear

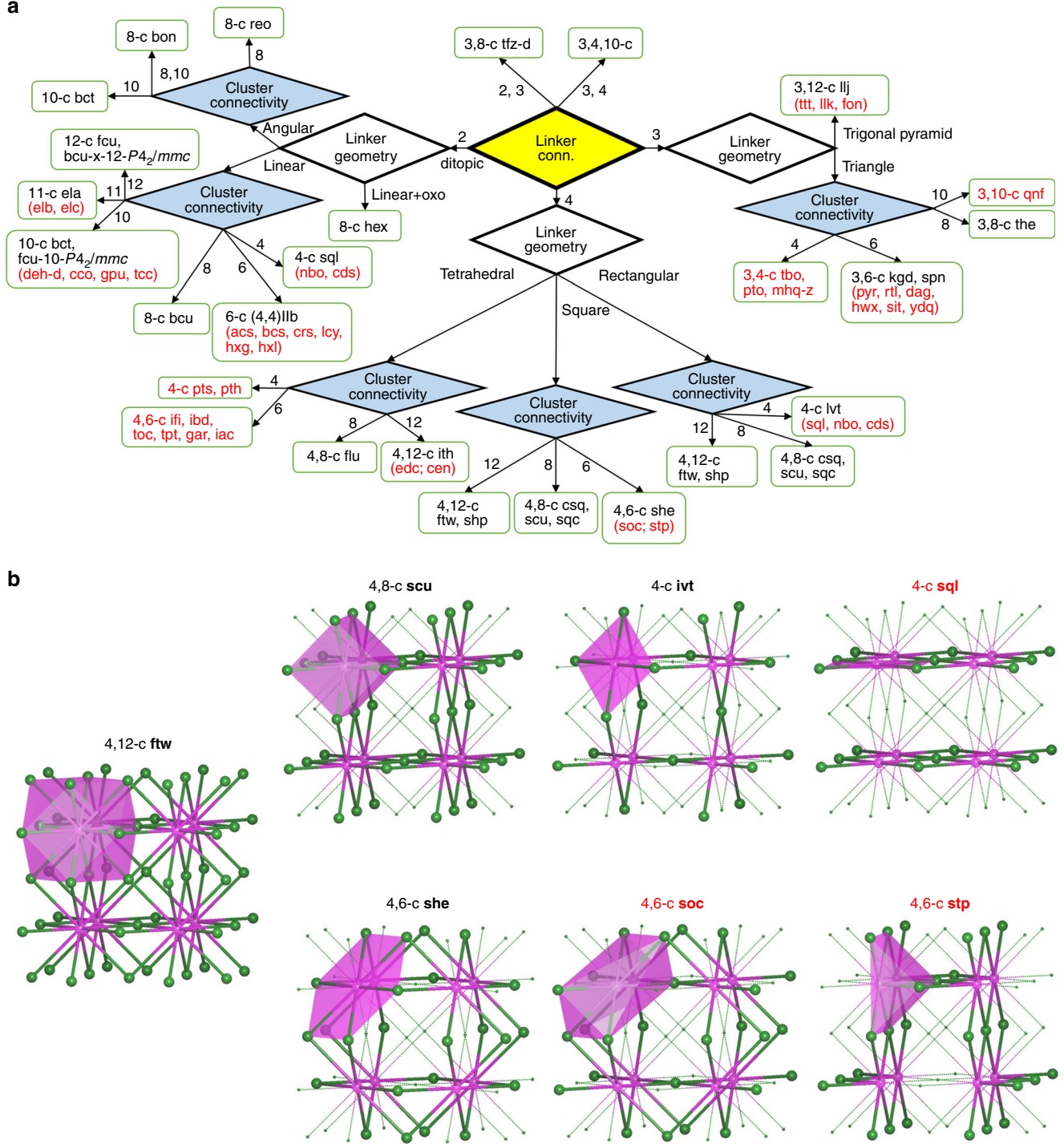

**Fig. 2** Topology analysis of Zr-based metal-organic frameworks (Zr-MOFs). **a** Topologies of Zr-MOFs built on $Zr_6$ clusters with ditopic (145) and polytopic (56) linkers. Structure types with minimum transitivity yet to be realized are shown in red. Numbers next to the arrows represent connectivity of the linker or cluster. **b** ftw net and its subnets with minimum transitivity (magenta and green balls represent the nodes of $Zr_6$ clusters and linkers, respectively). Structure types yet to be realized are shown in red

$M_2(COO)_4$ (M = Zn, Cu, etc.) SBU commonly observed for Cu or Zn based MOFs[44]. The resulting 3D structure adopts a rarely reported 4-c **lvt** topology[45].

**Topological analysis**. Topology-directed design of MOFs (or reticular chemistry)[46] has proven to be a powerful tool in creating various structure types for specific applications, as exemplified by MOFs constructed from $Zn_4O(COO)_6$ clusters and ditopic

ligands[47]. Zr-MOFs built on tetratopic linkers exhibit tremendous diversity in their structure topology, depending on the geometry, symmetry, and connectivity of both inorganic clusters and organic linkers. Noticeably, the organic linker chosen to construct a framework plays a vital role in the resulting underlying topology of this subgroup of MOFs. We have carried out a detailed analysis for 211 Zr-MOFs constructed from $Zr_6$ cluster and ditopic or polytopic linkers reported so far, with special attention to 44 Zr-

MOFs with tetratopic linkers using ToposPro approach and summarized our findings in Fig. 2[48,49], Supplementary Tables 4–6, and Supplementary Data 7. Out of these 44 Zr-MOFs, a total number of five structures are constructed from tetrahedron shaped ligands. The underlying topology of the remaining 39 structures built on planar tetratopic linkers possess 4,12-c **ftw/shp**, 4,8-c **csq/sqc/scu**, and 4,6-c **she** for 12-, 8-, and 6-connected $Zr_6$ clusters, respectively. The **ftw** topology is recognized to be the most thermodynamically stable structure type and has the largest porosity and lowest propensity for framework catenation making it desirable for adsorption related applications. Furthermore, a closer look at the 18 **ftw** structures reported to date shows that 14 of them demonstrate the same symmetry as the idealized net, even when the ligand has a rectangular shape. This reflects in a disorder of the ligands on the two possible relative orientations giving an average structure as 3,12-c **xxv** net which is observed for all structures containing a porphyrinic ligand (Supplementary Fig. 7). The four least symmetric **ftw** structures have rectangular ligands and two of them have an aspect ratio greater than one (1.03 and 1.15), deviating from the hypothesis that an **ftw** structure must use a ligand with unitary aspect ratio[34,35]. In particular, the bptc$^{4-}$ linker in compound **1** has the largest aspect ratio of 1.45 that gives the ordered **ftw** related net 3,12-c **kle**. The only other known example of **kle** net is NU-1000, which incorporates a pyrene-based tetratopic ligand with an aspect ratio of 1.03[50]. These examples suggest that with proper synthetic conditions, other **ftw**-related **kle** structures may be possible with ligands of aspect ratio greater than 1.0. However, it should be noted that, while a square shaped tetratopic ligand is not a necessary requirement for the formation of **ftw** topology, there is an upper limit in the ligand aspect ratio above which such a topology will no longer be thermodynamically favored. This is indeed the case for compounds **2** and **3**. The high aspect ratio of abtc$^{4-}$ (1.78) creates severe steric hindrance which prevents the formation of 12-connected $Zr_6$ SBU. As a result, compound **2** adopts the 4,8-c **scu** topology. The **scu** is by itself a unique net, with very few structures reported[31,40,42,43]. The higher aspect ratio of abtc$^{4-}$ also causes the distortion of the 8-connected $Zr_6$ cluster in compound **2**, and consequently a lower symmetry of $D_{4h}$, compared to $O_h$ symmetry of the 12-connected $Zr_6$ cluster in compound **1**. Having tptc-(Me)$_2^{4-}$, a ligand with an even higher aspect ratio of 2.28 incorporated in the structure, compound **3** adopts 4-connected $Zr_6$ clusters. The structure is a unique type and also represents the first example of a 3D Zr-MOF with a 4-c node $Zr_6$ cluster. Such coordination of a $Zr_6$ cluster has only been observed for ditopic ligands in a **sql** 2D layered MOF compound[51]. These results suggest ligand geometry plays an important role in determining the symmetry and connectivity of the $Zr_6$ SBU as well as the resulting topology of the Zr-MOF. This is shown by the scheme of the general relations between ligand connectivity and geometry, cluster connectivity, and resulting overall topology (Fig. 2a)[34,35,52]. It should be noted that each underlying net for giving connectivity of the ligand and cluster has the minimal possible transitivity. Transitivity is a measure of regularity of a net and the structure of MOFs follow a general principle that their underlying nets tend to be those of minimal transitivity[36]. Thus, 4,12-c **ftw/shp**, 4,8-c **csq/sqc/scu**, 4,6-c **she** contain two distinct nodes and only one type of edge (transitivity 21) and 4-c **lvt** is uninodal edge-transitive net (transitivity 11). Other useful relations can be derived by net–subnet approach[53]. The nets with lower coordinations such as 4,8-c **scu**, 4,6-c **she**, and 4-c **lvt** can be obtained from the 4,12-c **ftw** net (having the highest coordination) by removing 4-c nodes and hence reducing the ligand/cluster ratio. The relations together with the minimal transitivity principle provides the ground for predicting possible topological types of new Zr-MOFs yet to be realized, which are

subnets of the **ftw** net with minimal transitivity (Fig. 2b): 4,6-c **soc** (transitivity 21), 4,6-c **stp** (transitivity 21), and 4-c **sql** (transitivity 11)[36].

**Porosity and stability**. Porosity of compounds **1** and **2** has been confirmed by nitrogen adsorption measurements at 77 K. The Type I adsorption isotherm profiles indicate their microporous nature. The BET surface areas and micropore volumes are 1030 and 1318 m$^2$ g$^{-1}$, and 0.38 and 0.45 cm$^3$ g$^{-1}$ for compounds **1** and **2**, respectively (Supplementary Figs. 8 and 9). These values are higher than that of zeolite 5 A (BET surface area: ~600 m$^2$ g$^{-1}$, pore volume: 0.25 cm$^3$ g$^{-1}$). Remarkably, compound **1** also possesses exceptionally-high thermal and water stability (Fig. 3a and Supplementary Figs. 10–13). It is thermally stable up to at least 400 °C and the structure remains intact after being heated at 180 °C in open air for 1 month. It can also be immersed in aqueous solutions of pH = 2 to 12 for 1 week without losing any crystallinity. A close comparison is made between compound **1** and UiO-67 with respect to framework stability since both are made of the same 12-connected $Zr_6$ SBU and organic ligands with similar length[54]. Strikingly, compound **1** is much more robust than UiO-67 (Supplementary Figs. 14 and 15). After soaking in water at 80 °C for 1 day, both crystallinity and porosity of compound **1** were well retained whereas UiO-67 lost most of its long-range order and ~80% of its porosity (Supplementary Table 7). We attribute this to the difference in their ligand connectivity (4 in compound **1** and 2 in UiO-67). This observation also agrees with a previous computational study which demonstrates that MOFs built on polytopic ligands show higher stability than those made of ditopic ligands[55].

We further compare the stability of the three compounds synthesized in this work. Compound **2** exhibits good thermal and water stability (Fig. 3b, c, and Supplementary Fig. 16); however, it is not as robust as compound **1**. Although it retains crystallinity after various thermal/water treatments, it experiences a slight decrease in porosity after hot water treatment. Compound **3**, on the other hand, exhibits poor stability and suffers structural collapse upon thermal activation and shows almost no porosity (Supplementary Figs. 17–19). Based on the above analysis on their structures and topology, it is clear that the distinct difference in their framework stability correlates to the connectivity of their SBUs. SBUs with higher connectivity will lead to more robust frameworks. This suggests the relatively high stability of Zr-MOFs is not solely originated from strong Zr–O bonds. Other structural factors, including the geometry and connectivity of both SBU and ligand, all contribute to the overall robustness of the compounds. Additionally, ligand expansion usually results in reticular structures with reduced stability as seen in the case of the UiO-66/67/68 series[56].

**Single component adsorption of C6 isomers**. C6 isomers are selected to test the separation performance of compound **1** based on the consideration of its pore structure. Single component equilibrium isotherms of nHEX were collected on compound **1** at various temperatures (Supplementary Figs. 20 and 21). At 150 °C, a temperature chosen based on the industrial operation temperature range (100–200 °C), compound **1** takes up 130 mg g$^{-1}$ nHEX under a partial pressure of 110 torr, which is ~70% higher than that of the benchmark material zeolite 5 A with an uptake of 77 mg g$^{-1}$ under the same condition (Fig. 4a–c). To the best of our knowledge, this uptake capacity represents the highest value of any porous material under similar conditions. The ideal pore structure in compound **1** may account for its high nHEX uptake at high temperature: the large cages guarantee the void space needed to accommodate a large amount of adsorbates while the

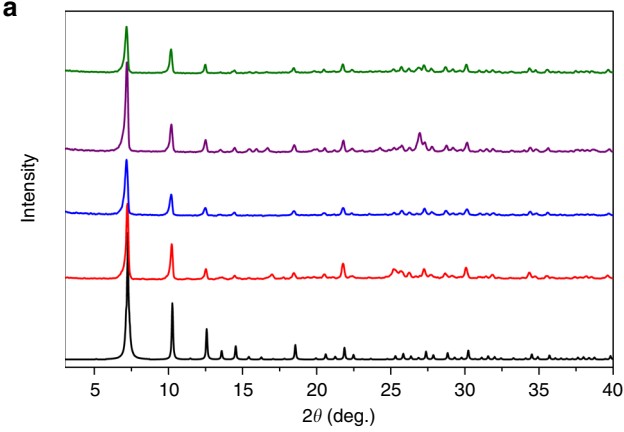

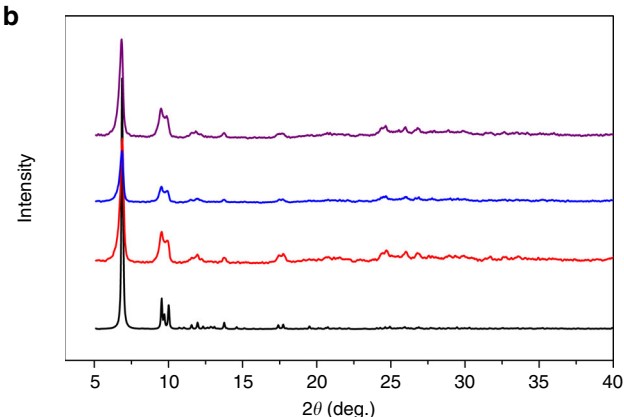

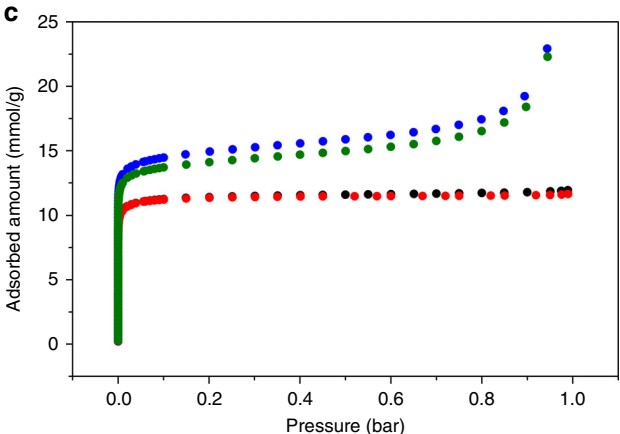

**Fig. 3** Stability test and porosity characterization of compounds **1** and **2**. **a** Powder X-ray diffraction patterns of compound **1**. From bottom to top: simulated (black), as synthesized (red), after adsorption experiments (blue), heated at 180 °C in open air for 1 month (purple, relative humidity: ~30–50% RH), and boiled in water for 1 week (green). **b** Powder X-ray diffraction patterns of compound **2**. From bottom to top: simulated (black), as synthesized (red), boiled in water for 3 days (blue), and heated at 180 °C in open air for 1 week (purple). **c** N$_2$ adsorption isotherms at 77 K for compound **1** (as synthesized: black, after being heated at 80 °C in water for 1 day: red) and compound **2** (as synthesized: blue, after being heated at 80 °C in water for 1 day: green)

suited pore aperture serves as a gate to control the diffusion of molecules into the cages. The strength of adsorbent–adsorbate interaction was evaluated by isosteric heat of adsorption ($Q_{st}$) calculated using adsorption isotherms at high temperatures (180, 200, 220 and 240 °C, Supplementary Fig. 22). A value of 48 kJ mol$^{-1}$

was obtained for nHEX. It is lower than that of zeolite 5 A (59 kJ mol$^{-1}$)[57] but higher than that of ZIF-8 (33 kJ mol$^{-1}$)[58]. Notably, adsorption kinetics of nHEX on compound **1** is comparable to that on zeolite 5 A at 150 °C and no diffusion restriction was observed (Fig. 4c). In contrast, compound **1** shows negligible adsorption of monobranched 3-methylpentane (3MP) and essentially no uptake of dibranched 2,3-dimethylbutane (23DMB, Supplementary Fig. 23). This is not surprising considering its small window size. The tiny amount of 3MP taken up by compound **1** may presumably be attributed to the surface adsorption or adsorption at defect sites[5]. This is further confirmed by breakthrough experiments and ab initio modeling, which will be discussed in the following sections. The selective adsorption of nHEX over the branched isomers by compound **1** follows a similar molecular exclusion mechanism as in the case of zeolite 5 A, but having a much higher uptake capacity. Compound **1** also exhibits excellent recyclability without losing any uptake capacity after ten adsorption–desorption cycles (Supplementary Fig. 24). Additionally, we have reproduced the best performing MOFs reported so far and evaluated and compared their adsorption capacity and selectivity for the separation of C6 alkane isomers (Supplementary Figs. 25–31). Clearly, compound **1** demonstrates both high adsorption capacity and selectivity under the test conditions, with a performance level comparable to zeolite 5 A.

While compound **1** represents a promising alternative to zeolite 5 A, compound **2**, on the other hand, may represent an attractive supplement rather than a substitute. As shown in Fig. 4d, e and Supplementary Figs. 32–36, all three hexane isomers can be accommodated into the channel of this compound owing to its larger pore size. In addition, it shows no diffusional limitations for linear or monobranched isomers at any temperature investigated with only slight restrictions for dibranched isomer at low temperature (Supplementary Fig. 37). The adsorption capacity of nHEX in compound **2** is ~105 mg g$^{-1}$ at 150 °C and 100 torr, which is slightly lower than that of compound **1** but higher than that of zeolite 5 A and most previously reported materials. Though the channels in compound **2** are large enough to adsorb all three isomers, it exhibits different extent of interaction with each individual isomer. As illustrated by the $Q_{st}$ calculations (Fig. 4f), nHEX is the most preferentially adsorbed species while the adsorption affinity for the dibranched isomer is the weakest. This can be explained by the degree of contact between the adsorbate and the channel surface: the linear hexane molecules can maximize its van der Waals interaction with the pore surface while the dibranched isomer is poorest as it is not flexible enough for sufficient contact with the channel[27].

**Column breakthrough measurements.** To mimic real-world conditions, it is important to carry out adsorption experiments with mixed adsorbates (Supplementary Fig. 38). To evaluate the capability of the title compounds for separating C6 alkane isomers under such conditions, we conducted column breakthrough experiments on both compounds **1** and **2** with an equimolar mixture of nHEX, 3MP and 23DMB at 150 °C. Measurements under identical experimental conditions were also performed on zeolite 5 A for comparison. The results are shown in Fig. 4g–i. Both compound **1** and zeolite 5 A adsorb nHEX exclusively. While branched C6 alkanes elute immediately, linear isomer shows a delayed retention. The real-time RON curves of the eluted product are also plotted in the figure. Before the breakthrough of nHEX, the RON values are higher than 90 for both adsorbent materials, meeting and exceeding the industrial standard for refined hexane blends (RON = 83). Under the same experimental conditions, nHEX breaks at the 59th minute on

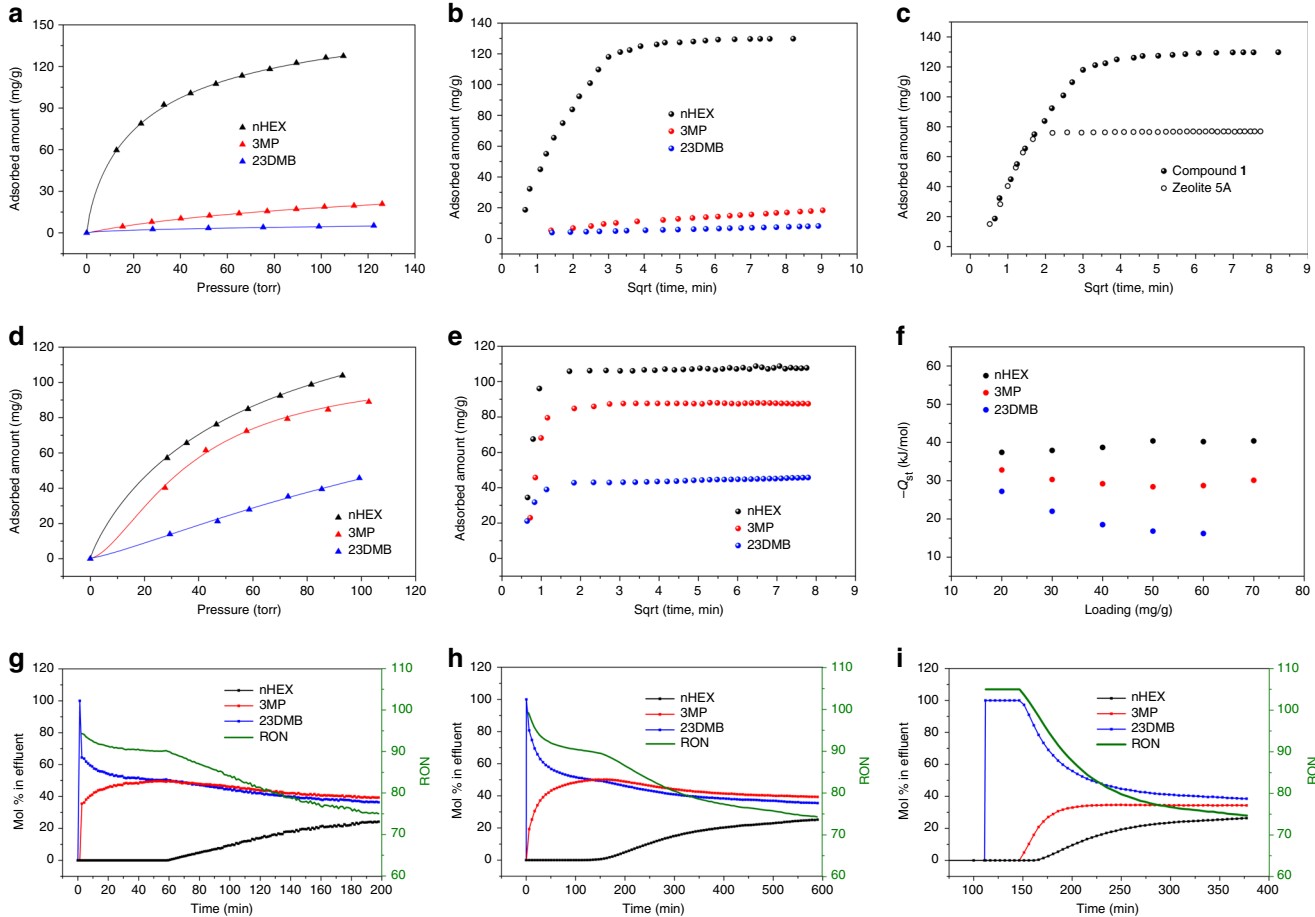

**Fig. 4** Adsorption and separation of C6 alkane isomers. **a** Adsorption isotherms and **b** adsorption rates (at 100 torr) for compound **1** at 150 °C. **c** *n*-Hexane adsorption rates for compound **1** and zeolite 5 A at 150 °C and 100 torr. **d** Adsorption isotherms and **e** adsorption rates (at 100 torr) for compound **2** at 150 °C. **f** Heat of adsorption of C6 alkane isomers on compound **2**. Breakthrough curves of an equimolar ternary mixture of C6 alkane isomers at 150 °C for **g** zeolite 5 A, **h** compound **1**, and **i** compound **2**. Green curve represents the real-time RON of the eluted products

zeolite 5 A and at the 118th minute on compound **1**, meaning the dynamic adsorption capacity (before breakthrough) of the latter is twice of that of the former. This is a significant improvement considering the fact that compound **1** retains the merit of complete exclusion of the branched isomers.

As stated above, branched alkanes (both monobranched and dibranched) break immediately from the column of zeolite 5 A or compound **1**, suggesting neither of these two materials is able to separate isomers of different degrees of branching. In contrast, compound **2** shows clean separation of monobranched and dibranched hexane isomers. As shown in Fig. 4i, breakthrough results for compound **2** indicate that 23DMB elutes first from the column. 3MP elutes at a much later time which is followed by nHEX. The breakthrough times are 112, 151, and 166 min for the dibranched, monobranched, and linear isomer, respectively. The significantly longer breakthrough time of 3MP than 23DMB indicates a good separation between mono- and di-branched isomers. The separation factor achieved for this compound (~1.3) is higher than Fe$_2$(BDP)$_3$ (~1.1), which represents the best MOF material for such a separation process prior to this work.[27] The steepness of the breakthrough events for all three isomers suggests that there are no diffusion restrictions and the separation is thermodynamically controlled. This is consistent with the single component adsorption results where compound **2** shows equilibrium adsorption toward all isomers at 150 °C but with different uptake amount and adsorption affinity. Notably, at the beginning of the breakthrough experiment, the eluted product has

a RON higher than 100, well above the value for the state-of-the-art benchmark material zeolite 5 A. This can be attributed to the material's ability to separate monobranched and dibranched alkane isomers, making it possible to obtain pure dibranched isomer with the highest RON value. This is of high importance for the petroleum refinement industry as it offers a method to further improve the quality of commercial gasoline. Breakthrough measurements at 30 °C show that the separation ability is retained for all compounds with higher uptake capacity and more diffusion restrictions (Supplementary Figs. 39–41).

**Computational modeling**. The experimental data from the previous sections clearly show that the selectivity of compound **2** is thermodynamically driven, as is expected due to the relatively large channels of this structure. On the other hand, for compound **1** with much smaller openings the data suggest a mechanism based on selective molecular exclusion and the necessary further details can be gained through computational modeling. To this end, we performed density functional theory (DFT) calculations, including ab initio molecular dynamics (AIMD), to model the kinetic diffusion barrier exerted on nHEX, 3MP, and 23DMB while passing through the pore window into the MOF cages. For details see the "Computational modeling subsection under Methods and Supplementary Note 1. AIMD calculations were necessary as the diffusion of guest molecules into the MOF framework is greatly aided by temperature. Diffusion events over

large barriers are statistically rare events on the timescale accessible through AIMD calculations, preventing a direct assessment of the diffusion barrier. We thus estimate the diffusion barrier as the difference in total energy of the isomer inside the MOF pore window and the isomer at the entrance (just outside the MOF pore window), as depicted in Supplementary Fig. 42; both AIMD runs were performed independently of each other.

Our original idea for diffusion of isomer molecules into the MOF was through the straight or diagonal entrances depicted in Supplementary Fig. 43f and g. However, in both models the ground-state energy barriers, i.e., the barriers corresponding to an optimized zero-temperature structure and transition-state search, for nHEX of more than 350 kJ/mol are too high to explain adsorption. We thus investigated the effects of temperature through a breathing mechanism where the adjacent organic linkers slightly open the pore windows in a breathing motion (Supplementary Fig. 43h). Breathing modes of pores in MOFs have been well characterized[59] and are naturally present at finite temperature[60], but are often also associated with external stimuli such as ion introduction[61] or added pressure[62]. A drastic drop-off in the ground-state barrier for nHEX is observed for the breathing entrance pathways compared to the straight or diagonal entrances (Supplementary Table 8). While the barrier did lower substantially using this breathing mechanism, it is still very high.

The high barriers suggest that a full treatment of temperature effects is necessary. Indeed, performing AIMD calculations to estimate the kinetic barriers at the experimental temperature of 150 °C lowers the barriers sufficiently to explain the experimentally observed adsorption—see Fig. 5a and Supplementary Fig. 44. From the latter, we can directly learn about the time-dependent energy fluctuations and the resulting change in kinetic energy barrier. nHEX has a time-averaged barrier of 92 kJ/mol while the values for 3MP and 23DMB are well above 200 kJ/mol (Fig. 5a and Supplementary Table 9). An energy barrier of 100 kJ/mol is approximately an upper limit to have non-negligible statistical probabilities for molecules to pass through the pore window at these temperatures. The AIMD results indicate that the energetic barrier for nHEX is below 100 kJ/mol for 59% of the time. In contrast, the energy for branched isomers never drops below 100 kJ/mol during that time period, which suggests that their diffusion into the MOF cages is effectively suppressed. Most noteworthy, however, is that not only the energy barrier for nHEX is less than 100 kJ/mol for much of the AIMD trajectory time, but there are brief periods of time where the energy barrier is less than zero (see Supplementary Fig. 44), indicating that it is energetically favorable for the adsorbate molecules to enter the cage. While these less-than-zero barrier times are brief, we estimate them to occur with a frequency of at least one per ps such that—over the experimental observation period—they help to explain the fast diffusion observed for the nHEX molecule. These results are in good agreement with the experimental results that nHEX experiences equilibrium adsorption while 3MP and 23DMB are excluded from the pores of compound **1**.

**Infrared spectroscopy**. To examine the interaction of C6 alkane isomers with compound **2** and gain insight into the thermodynamically-driven selective adsorption, we performed IR spectroscopy studies on the alkanes-loaded MOF (see "IR spectroscopy study under Methods for experimental details). Figure 5b shows the IR spectra of compound **2** upon adsorption of nHEX, 3MP, or 23DMB. For all three alkane gases, positive bands, located in the 3000–2850 cm$^{-1}$ and 1355–1470 cm$^{-1}$ regions, are associated with the C–H stretching and bending vibrations of the alkanes in the gas phase as they disappear quickly upon the evacuation of the cell (Supplementary Fig. 45).

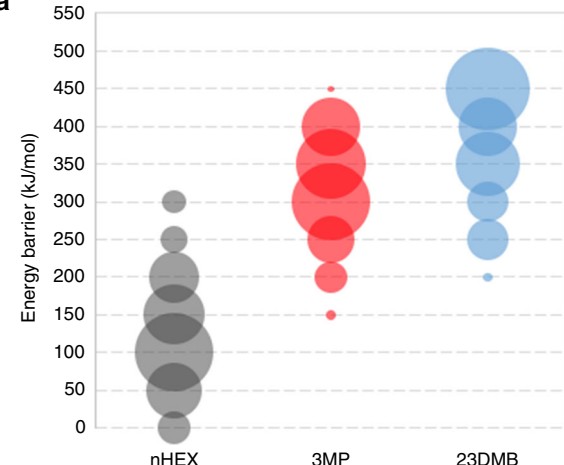

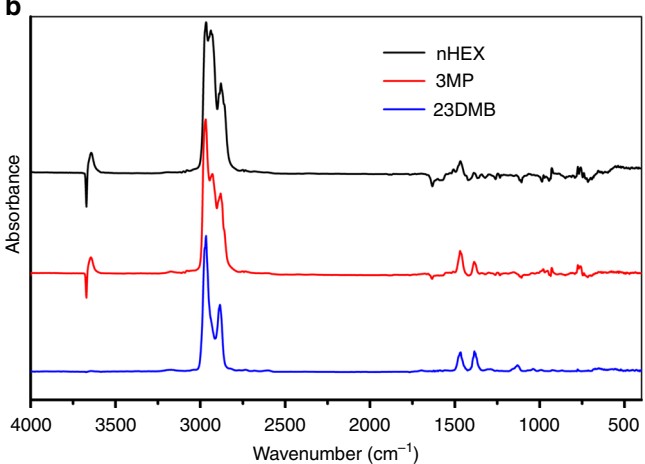

**Fig. 5** Investigation of the mechanism of separation of C6 alkane isomers in compounds **1** and **2**. **a** Energy barriers calculated from ab initio molecular dynamics (AIMD) simulations in compound **1** to highlight vast differences in kinetic energy barrier between the linear and branched C6 alkanes. Barriers range from 0 (no barrier) to 450 kJ/mol and are broken into 50 kJ/mol increments. The size of each circle corresponds to the amount of AIMD trajectory time in that increment. A larger circle thus indicates more AIMD time steps with a barrier in that increment range. As expected, a higher degree of branching results in higher barriers. These results confirm the kinetically-driven sieving mechanism at play in compound **1**. **b** IR spectra of compound **2** upon adsorption of three C6 alkane isomers after 3 min under 10 torr at 30 °C. All spectra are normalized by the reference spectra recorded prior to the adsorption

Therefore, only molecules adsorbed inside the MOF are detected through the corresponding vibrations that are typically shifted or modified as a result of their interactions with the framework. Interestingly, both nHEX and 3MP molecules exhibit distinct red shifts in the 3600 cm$^{-1}$ spectral region (from the initial position of 3671 cm$^{-1}$ to ~3643 cm$^{-1}$) and a loss at 1633 cm$^{-1}$. In contrast, there is no detectable shift in the 3600 cm$^{-1}$ spectral region and no loss at 1633 cm$^{-1}$ during the adsorption of 23DMB. The peak initially at 3671 cm$^{-1}$ is associated with the O–H stretch vibration of the terminal hydroxyl group in the SBU. A red shift therefore indicates that the adsorbed species are interacting with this group through weak hydrogen bonding[63]. We postulate that the loss of peak at 1633 cm$^{-1}$ observed during the adsorption of nHEX or 3MP is due to an interaction between the adsorbed molecules and the carboxylate groups of the organic linker abtc$^{4-}$

of the MOF structure. Altogether, the in situ IR measurements of compound **2** point to an interaction between nHEX (or 3MP) with the hydroxyl and carboxylate groups of the MOF, which is absent in the case of 23DMB. These findings are consistent with the results found in the previous sections for 23DMB: lower adsorbed amounts, weaker affinity (lower heat of adsorption), and much shorter retention time in column breakthrough measurements.

## Discussion

Efficient separation of hydrocarbons via selective adsorption by porous solids requires precise control of their pore structure. On the basis of reticular chemistry, metal-organic frameworks with tailored structures can be synthesized by judicious selection of inorganic clusters and organic linkers. This designer strategy, combined with the general high stability of Zr-MOFs, offers an opportunity to develop high performance adsorbent materials with desirable structures and optimal pore shape/size for targeted separation processes. In this study, we have analyzed the influence of the ligand size and geometry (aspect ratio) on the SBU connectivity and resulting framework structure, and on the framework stability of various topologies for Zr-MOFs. Guided by such analysis, we have achieved a new 4,12-c **ftw** structure with cage-like pores. Compound **1** possesses exceptionally high framework stability and optimized pore aperture. It selectively adsorbs nHEX but excludes its branched isomers, with an adsorption capacity that outperforms the commercial benchmark material, zeolite 5 A. Computational modeling confirms that the adsorption process follows a size exclusion (sieving) mechanism in which nHEX easily passes into the MOF cage through the MOF pore window while the branched isomers cannot. Column breakthrough experiments on a mixture of three C6 alkane isomers show that compound **1** exclusively adsorbs linear alkane, similarly to zeolite 5 A but with higher uptake capacity. A ligand with higher aspect ratio yields a 4,8-c **scu** structure (compound **2**) with channel-like pores. Compound **2** is capable of separating mono- and di-branched C6 alkane isomers (with RON >100) that is not achievable by the benchmark adsorbent zeolite 5 A. The 3MP/23DMB separation efficiency of compound **2** exceeds the best performing MOFs reported to date. The basis for compound **2**'s thermodynamically-controlled separation mechanism has been established by adsorption measurements, heats of adsorption calculations, and in situ IR spectroscopy studies. The findings offer a supplementary and readily implementable technology for the petroleum refinement industry to further improve the octane rating of commercial gasoline products.

## Methods

**Materials**. All reagents were used as received unless otherwise specified. Detailed syntheses of organic ligands are described in Supplementary Methods and Supplementary Figs. 46–48.

**Synthesis of compound 1**. Zirconium (IV) oxychloride octahydrate ($ZrOCl_2 \cdot 8H_2O$, 32.2 mg, 0.1 mmol) was ultrasonically dissolved in a mixed solvent of N,N-dimethylformamide (DMF, 5 mL) and formic acid (5 mL) in a 20 mL scintillation vial. $H_4$bptc (33 mg, 0.1 mmol) was then added to the solution which was sonicated for 5 min before being moved to a preheated oven at 120 °C. The reaction was kept at 120 °C for 3 days and microcrystalline white powder was obtained through centrifuge. The materials were washed with DMF and methanol with a Soxhlet extractor for 2 and 3 days, respectively, prior to the adsorption study. Yield: 68% (based on Zr). The same crystalline phase was obtained when the ratio for DMF:formic acid (v:v) is between 0 and 1.5 (total volume is 10 mL), beyond which the reaction generated an amorphous gel. Teflon lined vessels were used for all reactions when the ratio was less than 1 to prevent evaporation of the solvent.

**Synthesis of compound 2**. Zirconium (IV) oxychloride octahydrate ($ZrOCl_2 \cdot 8H_2O$, 32.2 mg, 0.1 mmol) was ultrasonically dissolved in a mixed solvent

of N,N-dimethylformamide (DMF, 8 mL) and formic acid (6 mL) in a 20 mL scintillation vial. $H_4$abtc (35.8 mg, 0.1 mmol) was then added to the solution which was sonicated for 5 min before being moved to a preheated oven at 120 °C. The reaction was kept at 120 °C for 3 days and light orange solids were observed in the reaction glass vial upon cooling. These solids are either microcrystalline or crystals large enough for single crystal X-ray diffraction analysis. The solid samples were collected by centrifuge (or vacuum filtration). The materials were washed with DMF and methanol with a Soxhlet extractor for 2 and 3 days, respectively, prior to the adsorption study. Yield: 74% (based on Zr). Compound **2** remained to be the only phase for the DMF:formic acid ratio (v:v) between 0 and 1.4 (total volume is 14 mL). Similar to the synthesis of compound **1**, an amorphous gel would form beyond this range. Teflon lined vessels were used for all reactions where the ratio was less than 1.

**Synthesis of compound 3**. Zirconium (IV) oxychloride octahydrate ($ZrOCl_2 \cdot 8H_2O$, 32.2 mg, 0.1 mmol) was ultrasonically dissolved in a mixed solvent of N,N-dimethylformamide (DMF, 8 mL) and formic acid (4.5 mL) in a 20 mL scintillation vial. $H_4$tptc-(Me)$_2$ (21.7 mg, 0.05 mmol) was then added to the solution which was sonicated for 5 min before being moved to a preheated oven at 120 °C. The reaction was kept at 120 °C for 3 days and colorless crystals were obtained through centrifuging. Yield: 55% (based on Zr). The same crystalline phase was obtained when the amount of formic acid was between 3.5 and 4.5 mL (8 mL DMF), beyond which either an amorphous gel (less acid) or a clear solution (more acid) was observed.

**Characterizations**. Nuclear magnetic resonance (NMR) data were collected on a 400 MHz Oxford NMR. Powder X-ray diffraction (PXRD) analysis was carried out on a Rigaku Ultima-IV automated diffraction system using Cu Kα radiation ($\lambda = 1.5406$ Å). The data were collected at room temperature in a $2\theta$ range of 3–40° with a scan speed of 2° min$^{-1}$. The operating power was 40 kV/40 mA. Thermogravimetric analyses (TGA) of samples were performed using the TA Instrument Q5000IR thermal gravimetric analyzer with nitrogen flow and sample purge rate at 10 ml min$^{-1}$ and 25 ml min$^{-1}$ respectively. About 3–5 mg of sample was loaded onto a platinum sample pan and heated from room temperature to 600 °C at a rate of 10 °C min$^{-1}$ under nitrogen flow. N$_2$ adsorption at 77 K was performed on a Micromeritics 3Flex adsorption analyzer. Prior to each measurement, ~100 mg of solvent exchanged sample was activated under dynamic vacuum overnight (300 °C for compound **1** and 150 °C for compounds **2** and **3**). Single-crystal synchrotron X-ray diffraction data of Compounds **2** and **3** were collected at 100 K on a D8 goniostat equipped with a Bruker PHOTON100 CMOS detector at Beamline 11.3.1 at the Advanced Light Source (ALS) in Lawrence Berkeley National Laboratory, using synchrotron radiation tuned to $\lambda = 0.7749$ Å. The structure was solved by direct methods and refined by full-matrix least-squares on F$^2$ using the Bruker SHELXTL package (Supplementary Notes 2 and 3).

**Crystal data for compound 1**. Cubic crystal system, space group *Im-3*, $a = 24.3597$ (3) Å, $V = 14455.0(5)$ Å$^3$, $Z = 8$. CCDC No.: 1557153.

**Crystal data for compound 2**. Monoclinic crystal system, space group *C2/m*, $a = 25.4692(11)$ Å, $b = 36.3589(15)$ Å, $c = 21.5275(9)$ Å, $\beta = 122.260(2)°$, $V = 16857.8$ (13) Å$^3$, $Z = 8$. CCDC No.: 1557154.

**Crystal data for compound 3**. Orthorhombic crystal system, space group *Imma*, $a = 25.2379(14)$ Å, $b = 27.7529(16)$ Å, $c = 15.2344(10)$ Å, $V = 10670.6(11)$ Å$^3$, $Z = 8$. CCDC No.: 1557155.

**Hydrocarbon adsorption measurements**. Single component hydrocarbon adsorption measurements were performed on a homemade gravimetric adsorption analyzer modified from a TGA Q50 (TA Instruments). Ultra-high pure N$_2$ (99.999%) was used as a carrier gas passing through a bubbler filled with liquid hydrocarbon. The partial pressure of hydrocarbon was controlled by adjusting the blend ratio of pure N$_2$ and N$_2$ saturated with hydrocarbon vapor. Adsorbed amount was monitored by the weight change of the sample. For a typical measurement, ~20 mg sample was first activated at its outgassing temperature (300 °C for compound **1** and 180 °C for compound **2**) under pure N$_2$ flow for 2 h to remove the initial solvent residing in the pores. The temperature was then cooled to adsorption temperature and hydrocarbon vapor was introduced into the adsorption chamber. Mass of the adsorbent was recorded throughout the experiment.

**Column breakthrough experiments**. Multicomponent breakthrough experiments were conducted using a lab-scale fix-bed packed with the adsorbent sample. For a typical measurement, 1.0 g of activated sample was packed into a quartz column (5.8 mm I.D. × 150 mm) with silane treated glass wool filling the void space. Pure N$_2$ flow was used for initial purging of the adsorbent and then N$_2$ saturated with an equimolar mixture of all three C6 alkane isomers was passed through the column. The effluent from the column was monitored using an online GC equipped with HP-PONA column and FID.

**Computational modeling**. Ab initio modeling was performed on compound **1** at the density functional theory (DFT) level with VASP[64,65]. The standard VASP PAW pseudopotentials were implemented with an energy cutoff of 600 eV, and only the Γ point was used. Due to the size of compound **1**, appropriate cutouts of the MOF were taken directly from the experimental CIF file to perform the DFT calculations. In order to find temperature dependent energy barriers, ab initio molecular dynamics (AIMD) calculations using a Verlet algorithm were performed; energy barriers were then calculated via appropriate energy differences as explained in the text. The AIMD trajectories were run for over 1.5 ps and data was collected after a 200 fs thermalization period. Further details are available in Supplementary Note 1.

**IR spectroscopy study**. For in situ IR absorption measurements of the C6 isomers in compound **2**, 2 mg of the MOF powder was pressed onto a tungsten mesh (diameter of 1.2 cm and 1 mm thick). The sample was placed into an environmental cell at the focal point of the sample compartment of the infrared spectrometer. The cell was connected to a vacuum line for evacuation (base pressure of 20 mTorr). The samples were activated under vacuum (20 mTorr) at 150 °C until the IR spectra did not show any more changes (complete loss of water vapor). A pressure of 10 Torr was established for each of the dry C6 isomers vapors (nHEX, 3MP, or 23DMB) in the cell and all spectra were recorded in transmission at 30 °C between 400 and 4000 cm$^{-1}$ (4 cm$^{-1}$ spectral resolution, sufficient for the inhomogeneous widths of >10 cm$^{-1}$). The gas evacuation and desorption from the MOF were monitored as a function of time after pumping was initiated.

**Data availability**. The X-ray crystallographic data for crystal structures reported in this study are provided as cif files in Supplementary Data 1–6 and have been deposited at the Cambridge Crystallographic Data Center. These data can be obtained free of charge from the Cambridge Crystallographic Data Centre via www.ccdc.cam.ac.uk/data_request/cif, with CCDC codes 1557153, 1557154, and 1557155. All references, topologies and geometrical parameters for the 145 + 56 structures listed on Supplementary Tables 4–6 and Fig. 2a are reported in the Supplementary Data 7. All other data, if not included in the Article or the Supplementary Information, are available from the authors on request.

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

## Acknowledgements

We would like to thank the financial support from the Materials Sciences and Engineering Division, Office of Basic Research Energy Sciences of the U.S. Department of Energy through Grant no. DE-FG02-08ER-46491. Y.H. acknowledges the KAUST CCF fund for supporting this study. The Advanced Light Source is supported by the Director, Office of Science, Office of Basic Energy Sciences, of the U.S. Department of Energy under Contract No. DE-AC02-05CH11231. T.T. acknowledges generous support from the Simons Foundation through Grant no. 391888, which endowed his sabbatical at MIT. D.M.P. thanks the Russian Government (Grant 14.B25.31.0005). E.V.A. is grateful to the Russian Science Foundation (Grant no. 16-13-10158). The RU team would also like to acknowledge Micromeritics Instrument Corp. for the award of a 3Flex system through its Instrument Grant program. H.W. would like to thank Ever Velasco for helpful discussions.

## Author contributions

J. Li and H.W. conceived and designed the research, and co-wrote the manuscript. H.W., Q.X., and Q.W. carried out the ligand design, materials synthesis and characterization. H. W. and D.H.O. performed the single component adsorption experiments. X.D. and Y.H. conducted column breakthrough measurements. J. Lin, J.S., and S.J.T. carried out X-ray diffraction analysis and determined the crystal structures. J.S. and T.T. performed theoretical calculations. J.C., K.T. and Y.J.C. collected and analyzed IR data. E.V.A. and D.M. P. did topology analysis. All authors contributed to the discussion of results and commented on the manuscript.

## Additional information

**Competing interests:** The authors declare no competing interests.

