## [Peer Review File · Nature Communications]

Reviewers' comments:

Reviewer #1 (Remarks to the Author):

Title: Topologically Guided Tuning of Zr-MOF Pore Structures for Highly Selective Separation of C6 Alkane Isomers

Nature Communications manuscript NCOMMS-17-20800-T

In this work, the authors show that with topologically directed approach a series of highly robust Zr-based MOF materials have been achieved which show nice performance in the separation of C6 alkane isomers. Overall this work is of high quality in light of the following points: a) topology directed synthesis/reticular chemistry is not new in the design/synthesis of MOFs/COFs materials, however, it's interesting in this work the authors are reporting a series of MOF structures that build on similar inorganic clusters but have different connectivity (not isorecticular) as a result of the different aspect ratios of the organic linkers used. b) The authors have done a thorough summary/analysis on the topology of reported Zr-MOFs, starting from the three new compounds obtained in this work. Rarely, topology-stability relations have also been explored. The results could help those who are striving for stable MOF materials for certain applications. c) These materials show impressive performance for the separation of alkane isomers: Compound 1 surpasses the state of the art material zeolite 5A under relevant conditions while Compound 2 demonstrates good separation of mono- and di-branched alkanes which has been a challenging task.

This is a solid work and represents a significant progress in the fields of MOFs design/synthesis and adsorptive separation of hydrocarbons. Therefore I recommend its publication on Nature Communications, but after minor revisions by addressing the issues/comments below:

- 1) The authors claim the MOF structures (topology, cluster connectivity) are mainly determined by the aspect ratio of the organic linker, it would be interesting to know whether the amount of modulators used during synthesis could also influences the topology.
- 2) The authors should include yield in MOF synthesis.
- 3) It shows Compound 1 maintained its structure after being heated at 180C in open air for one month. The authors should indicate the humidity level during the test.
- 4) Though Compound 3 is not robust toward activation, as pointed out by the authors, it did show some adsorption toward N₂ (although the porosity is much lower than the other compounds). How far is the measured porosity from the value estimated with its crystal structure?
- 5) Each supplementary figure should be cited in the main text. For example, the authors discussed the stability of Compound 1 at high temperature or wide pH values in Line 241-243 without citing any data sources/figures (Figure S11 and S12?). In addition, the supplementary materials should be reordered to improve the readability.
- 6) The pressure unit in Figure S26 is bar?

Reviewer #2 (Remarks to the Author):

This is a fine paper that can be accepted with minor modifications:

Comments:

The first(?) example of a scu-MOF of was reported in: DOI: 10.1039/B605762D (Communication) CrystEngComm, 2006, 8, 666-669.

This needs to be clarified: "This reflects in a disorder of the ligands on the two possible relative orientations giving an average structure as 3,12-c xxv net which is observed for all structures containing a porphyrinic ligand."

Supp. Mat:

What is the solvent for the NMR?

Many H-NMR peaks reported for H4abtc, is that correct?

Elemental analysis for H4tptc-2Me and other new compounds?

Comp 2: "light orange crystals were obtained through centrifuge." Please elaborate.

Refs. missing for EXPO and TOPAS methods

Figure S42. Zr-bptc is compound 1?

Reviewer #3 (Remarks to the Author):

I found the writing for the most to be unusually clear and well polished. In addition the introduction did an excellent job of putting the work in context. The main results are impressive and are well described.

The biggest problem I found was the description of the computational methods and results. The authors do not say what they did (DFT calculations?). And their description is very loose; for example "simulations" are not explained. They discuss temperature-dependent barrier. What precisely do they mean by that? The Born-Oppenheimer potential energy surface is independent of temperature. The computational results are described in a woefully inadequate way; it is so based that I cannot even begin to list all the problems. And I am sorry to say that it makes one suspect that they do not know what they are doing.

The authors should give a reference explaining the ftw structure at the place where it is first mentioned on page 5.

On page 10, the authors discuss steric hindrance as a possible thermodynamic impediment to synthesizing a structure. The synthesis procedure is more complicated though, and there could also be kinetic considerations.

Page 11: What is the difference between "permanent porosity" and just plain "porosity"? Does this refer to hot water treatment? This needs to be stated more clearly.

The table of contents of the supporting information is wrong. For example, it says the computational section begins on page 56 but actually it begins on 59.

Reviewer #4 (Remarks to the Author):

Crystallographic review.

Three new structures of zirconium MOFs are reported, one refined from powder X-ray data using the Rietveld method and two refined from single crystal X-ray data. The important information extracted from these structures with respect to this work is the identity, conformation and connectivity of the ligands. In all three structure determinations, the experimental data and analyses adequately support the conclusions drawn by the authors in respect to these key points. Determining the structures of highly porous and somewhat flexible materials is a difficult crystallographic challenge, further complicated in these examples by problems with small crystals, twinning and diffuse diffraction. Use of synchrotron radiation in the single crystal determinations has allowed the highest possible quality data to be measured. The structure refinements have obviously been challenging, given the diffraction quality of the crystals. The authors have made good use of the 'refinement_special_details' and 'validation response form' sections of the CIF to describe and explain the work. I recommend the crystallographic work in this manuscript is fit for publication, subject to the authors acting on or responding to the following points:

Structure 1

A more detailed description of the geometric restraints and constraints used in the Rietveld refinement should be included in SI.

Structure 2

The asymmetric unit contains two complete ligand units and as such an opportunity exists to improve the quality of the structure model by including geometric similarity restraints (e.g. SADI) to chemically similar bond distances e.g. N1-N2 and N3-N4.

The blanket use of ISOR restraints with a very small esd on all ellipsoids in the structure is highly inappropriate. The restraint should be used more sparingly with larger 'softer' esds and in conjunction with more physically realistic rigid bond and similarity restraints e.g. RIGU and SIMU. Large electron density peaks of up to $3 \text{ e } \text{Å}^{-3}$ occur close to the Zr₆ metal cluster – a comment on their origin and significance should be added.

The well written and useful contents of the 'refinement_special_details' and 'validation responses' from the CIF should be included in SI.

Structure 3

Again, blanket use of ISOR restraints is inappropriate and an effort should be made to substitute them for more appropriate restraints.

The structure has been Squeezed to treat disordered voids regions: the output of the Squeeze routine should be appended to the CIF and an effort made to identify the void contents based on the estimated electron count and void volume.

A final comment on the interpretation of all three crystal structures relates to the computational aspect of this work. Where the atoms coordinates from the CIFs have been used to create local fragments for computational calculations, consideration should be made as to whether the structures need to be optimised using theory prior to use. Whilst the structures presented are adequate for confirmation of connectivity and conformation, the large uncertainties on parameters and irregular bond lengths highlight that this is not a precise interrogation of atomic geometry.

RESPONSE TO REVIEWERS:

Reviewer #1 (Remarks to the Author):

Title: Topologically Guided Tuning of Zr-MOF Pore Structures for Highly Selective Separation of C6 Alkane Isomers

Nature Communications manuscript NCOMMS-17-20800-T

In this work, the authors show that with topologically directed approach a series of highly robust Zr-based MOF materials have been achieved which show nice performance in the separation of C6 alkane isomers. Overall this work is of high quality in light of the following points: a) topology directed synthesis/reticular chemistry is not new in the design/synthesis of MOFs/COFs materials, however, it's interesting in this work the authors are reporting a series of MOF structures that build on similar inorganic clusters but have different connectivity (not isorecticular) as a result of the different aspect ratios of the organic linkers used. b) The authors have done a thorough summary/analysis on the topology of reported Zr-MOFs, starting from the three new compounds obtained in this work. Rarely, topology-stability relations have also been explored. The results could help those who are striving for stable MOF materials for certain applications. c) These materials show impressive performance for the separation of alkane isomers: Compound 1 surpasses the state of the art material zeolite 5A under relevant conditions while Compound 2 demonstrates good separation of mono- and di-branched alkanes which has been a challenging task.

This is a solid work and represents a significant progress in the fields of MOFs design/synthesis and adsorptive separation of hydrocarbons. Therefore I recommend its publication on Nature Communications, but after minor revisions by addressing the issues/comments below:

Response: We thank this reviewer for the positive comments.

1) The authors claim the MOF structures (topology, cluster connectivity) are mainly determined by the aspect ratio of the organic linker, it would be interesting to know whether the amount of modulators used during synthesis could also influences the topology.

Response: For the material design and synthesis part of this study, we focused primarily on the investigation of relations between the aspect ratio of the tetratopic linkers and the connectivity/topology of the Zr-MOFs. But we agree fully with the reviewer that the amount of acid modulator is often a very important factor for the synthesis of Zr-MOFs. We have carried out a series of reactions with varying amount of acid modulators for the synthesis of each MOF and the results are summarized in Methods section (main text, pg. 21). It appears targeted compound remained to be the only crystalline phase with the selected range of modulator amount.

2) The authors should include yield in MOF synthesis.

Response: The yield for each compound has been added to Methods section in the main text (pg. 21).

3) It shows Compound 1 maintained its structure after being heated at 180C in open air for one month. The authors should indicate the humidity level during the test.

Response: The relative humidity level for the stability test was ~30-50 %RH. This information has been added to Figure 3 captions (revised text, pg. 34)

4) Though Compound 3 is not robust toward activation, as pointed out by the authors, it did show some adsorption toward N₂ (although the porosity is much lower than the other compounds). How far is the measured porosity from the value estimated with its crystal structure?

Response: The calculated surface area from the crystal structure of compound **3** is ~2500 m²/g while the experimental value is two magnitudes lower (33.6 m²/g). The sample loses its crystallinity upon activation, a clear indication of poor framework stability. This information has been added to Supplementary Fig. 19 (SI, pg. 19).

5) Each supplementary figure should be cited in the main text. For example, the authors discussed the stability of Compound 1 at high temperature or wide pH values in Line 241-243 without citing any data sources/figures (Figure S11 and S12?). In addition, the supplementary materials should be reordered to improve the readability.

Response: We thank the reviewer for careful checking. We have now cited all supporting figures in the main text. The supplementary sections have also been reordered.

6) The pressure unit in Figure S26 is bar?

Response: The original Figure S26 is now Supplementary Fig. 21. The unit is torr. There is a mistake in Supplementary Fig. 20 (originally Figure S25) where the unit should be torr, not bar. This has been corrected.

Reviewer #2 (Remarks to the Author):

This is a fine paper that can be accepted with minor modifications:

Response: We appreciate the positive comments from this reviewer.

Comments:

The first(?) example of a scu-MOF of was reported in: DOI: 10.1039/B605762D (Communication) CrystEngComm, 2006, 8, 666-669.

Response: We thank the reviewer for the information. We have now cited this article (Ref. #40 in the main text).

This needs to be clarified: “This reflects in a disorder of the ligands on the two possible relative orientations giving an average structure as 3,12-c xxv net which is observed for all structures containing a porphyrinic ligand.”

Response: We have made clarification by including a new figure (Supplementary Fig. 7, SI, pg. 7) to show the two possible relative orientations of a rectangular ligand in 4,12-ftw topology.

Supp. Mat:

What is the solvent for the NMR?

Response: The solvent used in the NMR analysis was DMSO-d⁶. This information has been added to Supplementary Methods (SI, pgs. 65, 66, and 68) and Supplementary Figs. 46-48.

Many H-NMR peaks reported for H₄abtc, is that correct?

Response: We thank this reviewer for pointing out this issue. The synthesis of H₄abtc was carried out following the procedure below:

“5-nitroisophthalic acid (2.1 g, 0.01 mol), NaOH (3.2 g, 0.08 mol), zinc powder (2.6 g, 0.04 mol) were mixed in ethanol/H₂O (50/20 mL). The mixture was kept under refluxing for 12 hours before cooled to room temperature. Yellow solid was obtained through vacuum filtration which was then dissolved in 80 mL 1M NaOH solution. Upon filtration, the filtrate was acidified with 6 M HCl to get orange solid which was recrystallized from DMF.”

We initially used 0.8 g of NaOH and 1.3 g of zinc powder in the synthesis, as reported in the literature. However, the NMR of the product obtained (our original Figure S3, see below) didn't seem to match with the expected spectrum of H₄abtc, as pointed out by the reviewer. We noticed that the product was actually a mixture of H₄abtc and H₄aobtc (Figure R1).

H₄abtc

H₄aobtc

Figure R1. ^1H NMR of a mixture of H_4abtc and H_4aobtc .

However, the existence of H_4aobtc as an impurity didn't affect the synthesis of Zr-abtc (compound **2**), and no aobtc^{4-} was observed in the crystal structure of the MOF produced. This is because in the process of a solvothermal reaction, H_4aobtc will be reduced to H_4abtc (see Wang et al. Chem. Mater. 2008, 20, 3145-3152, Ref. #34 in SI).

We have now further examined the ligand synthesis conditions and found that the product is largely dependent on the amount of NaOH and zinc powder used in the reactions. By varying the amount of NaOH and zinc powder, pure H_4aobtc , or pure H_4abtc , or a mixture of both with a certain ratio can be obtained (basically the more NaOH and zinc powder was used, the less H_4aobtc was observed). They all led to Zr-abtc in similar yield via solvothermal reactions. In our revised manuscript, we have included the synthesis conditions that lead to pure H_4abtc and the corresponding ^1H NMR spectrum (Supplementary Fig. 47).

Elemental analysis for $\text{H}_4\text{tptc-2Me}$ and other new compounds?

Response: Since all the organic linkers including $\text{H}_4\text{tptc-2Me}$ were previously reported in the literature and they were synthesized based on the reported procedures (references cited accordingly in SI), they have only been characterized by ^1H NMR to confirm their purity. They were also confirmed by the MOF crystal structures.

Comp 2: “light orange crystals were obtained through centrifuge.” Please elaborate.

Response: During the synthesis of compound **2**, light orange solids were observed in the reaction glass vial upon cooling. These solids are either microcrystalline or crystals large enough for single crystal X-ray diffraction analysis. The solid samples were collected by centrifuge (or vacuum filtration). This has been elaborated in Synthesis of compound **2** (main text, Methods section, pg. 21).

Refs. missing for EXPO and TOPAS methods

Response: References have been added for EXPO and TOPAS methods (Refs. #1 and #2 in SI).

Figure S42. Zr-bptc is compound 1?

Response: Zr-bptc in Supplementary Fig. 31 (Figure S42 in the original version) refers to compound **1**. It has been replaced by “compound **1**” in the revised text.

Reviewer #3 (Remarks to the Author):

I found the writing for the most to be unusually clear and well polished. In addition the introduction did an excellent job of putting the work in context. The main results are impressive and are well described.

The biggest problem I found was the description of the computational methods and results. The authors do not say what they did (DFT calculations?). And there description is very loose; for example "simulations" are not explained.

Response: We thank the reviewer for pointing out that the level of theory and other details should have been presented immediately with the computational results. However, detailed information was (and is) located under the *Methods* section labeled “Computational modelling” on page 23 in the main text, answering the question this reviewer is asking and providing further detail.

In addition, Supplementary Note 1 in the supplementary materials (SI, pgs. 59-60) also provides further detailed information. Nonetheless, we do agree with the reviewer that some more details and a reference to where further information can be found are very helpful early on in the main manuscript. To this end, the content in the main text where the computational results are first presented on page 16 has been revised accordingly (main text, pg. 16, Paragraph 2). We have also further improved the remaining text in that section to clarify the methods and approaches used (main text, pgs. 16-17). In addition, more details have been added to the SI to clarify the level of theory and what calculations were performed (SI, pg. 59, Paragraph 1). The next paragraph (SI, pg. 59, Paragraph 2) also provides further details on how our calculations were performed and the role of temperature.

We also agree with the reviewer that the term “simulation” may not have been the best choice and we apologize for its use. Consequently, we have replaced the word “simulation(s)” throughout the manuscript and the SI.

They discuss temperature-dependent barrier. What precisely do they mean by that? The Born-Oppenheimer potential energy surface is independent of temperature.

Response: As the reviewer points out, for a frozen set of atomic positions R_i the Born-Oppenheimer potential energy surface is completely determined and independent of temperature. However, this is really only the zeroth order approximation. In reality, all atom positions R_i constantly change due to thermal fluctuations and the R_i 's become a function of time and temperature. In turn, the Born-Oppenheimer potential energy surface changes for each new set of R_i and the barriers change along with it, as e.g. depicted in Supplementary Fig. 44. The thermal fluctuations at finite temperatures cause a noticeable change in the barrier in general, and in its average value in particular; it follows that the barrier is now temperature dependent, see e.g. the difference between the 300 and 423 K calculations in Figure 5a in the main text. In other words, finite temperature allows the MOF atoms to fluctuate—some fluctuations are random, while others are more organized such as the breathing mode discussed in the main manuscript—which influences the barrier height to a varying degree depending on temperature.

In our computational work, we noticed early on that going beyond that zeroth order approximation of a fixed Born-Oppenheimer potential energy surface is going to be crucial to describe the diffusion of those molecules through the MOF framework. Only when including thermal fluctuations with the help of our *ab initio* molecular dynamics calculations were we able to properly include finite temperature effects and find that breathing motions of the MOF pore window explain the experimentally observed diffusion.

The computational results are described in a woefully inadequate way; it is so based that I cannot even begin to list all the problems. And I am sorry to say that it makes one suspect that they do not know what they are doing.

Response: The referee is entitled to his/or her opinion and we respect that. We do believe that the theory was adequately described in the original manuscript in the Methods section and in the SI, but we did make further efforts to clarify throughout the manuscript and the SI.

The authors should give a reference explaining the **ftw** structure at the place where it is first mentioned on page 5.

Response: As suggested by the reviewer, we have added a reference on pg. 5 (Ref. #32, main text).

On page 10, the authors discuss steric hindrance s a possible thermodynamic impediment to synthesizing a structure. The synthesis procedure is more complicated though, and there could also be kinetic considerations.

Response: We thank the reviewer for this comment. Indeed we fully agree that kinetics is an important factor influencing MOF formation. We noted that often a clear solution or an amorphous

gel (not long-range ordered structure) would result if insufficient reaction time was provided. For each compound, it required a certain reaction time to form a crystalline phase. Our discussions on page 10 focuses on the relations between ligand geometry/dimensions (or steric hindrance as referred to by the reviewer) and the thermodynamically stable phases obtained. Within a typical reaction time, we did observe a clear correlation between the ligand aspect ratio and the connectivity/topology of the MOF structures, which can also be seen from the ligand configuration in the MOF structure. For example, we noticed a distortion of $abtc^{4-}$ in the structure of compound **2** due to its high aspect ratio and clearly it is not such a good fit as $bptc^{4-}$ for **ftw** type structure. Experimentally, within the time scale we used for the synthesis (a couple of days to a week or so) we didn't observe an obvious kinetic effect on the final connectivity but we do think it's an interesting topic and is worthy of further exploration in our future work.

Page 11: What is the difference between "permanent porosity" and just plain "porosity"? Does this refer to hot water treatment? This needs to be stated more clearly.

Response: For MOFs, "permanent porosity" typically refers to a porous structure whose porosity will sustain upon removal of solvent/guest molecules. However to avoid any confusions, we have simply use "porosity" throughout the manuscript.

The table of contents of the supporting information is wrong. For example, it says the computational section begins on page 56 but actually it begins on 59.

Response: Corrections have been made accordingly.

Reviewer #4 (Remarks to the Author):

Crystallographic review.

Three new structures of zirconium MOFs are reported, one refined from powder X-ray data using the Rietveld method and two refined from single crystal X-ray data. The important information extracted from these structures with respect to this work is the identity, conformation and connectivity of the ligands. In all three structure determinations, the experimental data and analyses adequately support the conclusions drawn by the authors in respect to these key points. Determining the structures of highly porous and somewhat flexible materials is a difficult crystallographic challenge, further complicated in these examples by problems with small crystals, twinning and diffuse diffraction. Use of synchrotron radiation in the single crystal determinations has allowed the highest possible quality data to be measured. The structure refinements have obviously been challenging, given the diffraction quality of the crystals. The authors have made good use of the 'refinement_special_details' and 'validation response form' sections of the CIF to describe and explain the work. I recommend the crystallographic work in this manuscript is fit for publication, subject to the authors acting on or responding to the following points:

Structure 1

A more detailed description of the geometric restraints and constraints used in the Rietveld refinement should be included in SI.

Response: More details regarding the restraints and constraints have been added in Supplementary Table 1 (SI, pg. 49).

Structure 2

The asymmetric unit contains two complete ligand units and as such an opportunity exists to improve the quality of the structure model by including geometric similarity restraints (e.g. SADI) to chemically similar bond distances e.g N1-N2 and N3-N4.

The blanket use of ISOR restraints with a very small esd on all ellipsoids in the structure is highly inappropriate. The restraint should be used more sparingly with larger ‘softer’ esds and in conjunction with more physically realistic rigid bond and similarity restraints e.g. RIGU and SIMU.

Large electron density peaks of up to $3 \text{ e } \text{\AA}^{-3}$ occur close to the Zr₆ metal cluster – a comment on their origin and significance should be added.

The well written and useful contents of the ‘refinement_special_details’ and ‘validation responses’ from the CIF should be included in SI.

Response: Compound 2 SIMU and RIGU now replaces globally ISOR. However, ISOR has been used on a limited number of atoms which still had large isotropic displacement parameters. SADI have also been used in the molecule of the ligands around the N-N bonds. The 3 electron peaks close to the Zr atoms is either a result from the non-ideal treatment of the twinning or disordered that could not be modelled from the quality of the data set. More details have been added to Supplementary Note 2 in SI (pgs. 61-62).

‘refinement_special_details’ and ‘validation responses’ from the CIF are now included in SI (pgs. 61-62).

Structure 3

Again, blanket use of ISOR restraints is inappropriate and an effort should be made to substitute them for more appropriate restraints.

The structure has been Squeezed to treat disordered voids regions: the output of the Squeeze routine should be appended to the CIF and an effort made to identify the void contents based on the estimated electron count and void volume.

Response: SIMU and RIGU now replace globally ISOR. However, ISOR has been used on a limited number of atoms which still had large isotropic displacement parameters. The SQUEEZE output is automatically appended to the end of the CIF, however, the following has been added to _refine_special_details:

“SQUEEZE was used to remove the electron density in the porous that could not be modelled as solvent. SQUEEZE reported Solvent Accessible Volume of 7192 \AA^3 and Electrons Found in S.A.V. to be 1112. As there is a mixture of DMF and water used it was not possible to approximate the amount of each therefore no solvent was included in the chemical formula. Based on the SQUEEZE output there could be anywhere from around 5 molecules for DMF to around 100 of water molecules.” More details have been added to Supplementary Note 3 in SI (pgs. 63-64)

A final comment on the interpretation of all three crystal structures relates to the computational

aspect of this work. Where the atoms coordinates from the CIFs have been used to create local fragments for computational calculations, consideration should be made as to whether the structures need to be optimized using theory prior to use.

Response: We thank the reviewer for the useful comments and suggestions on the computational aspects, which allow us to better present our results. The related details have been added to the SI (page 59, Paragraph 1) to clarify various points and in particular that the cut outs were taken directly from the experimental CIF file and were optimized before calculating the barriers:

Whilst the structures presented are adequate for confirmation of connectivity and conformation, the large uncertainties on parameters and irregular bond lengths highlight that this is not a precise interrogation of atomic geometry.

Response: We agree that a cut out is not an exact replica of the extended atomic geometry, but it locally is a truthful representation of the MOF and in particular the MOF pore window we are trying to model. However, due to the size of the unit cell with near 850 atoms, at the ab initio level a cut out is the only feasible and well-accepted approach. We have modified the text—see the text changes made in answer to the previous question—to clarify that the cut outs were made directly from the experimental CIF file and how they were optimized.

We hope that these changes are satisfactory, and we thank the reviewers again for their their valuable and insightful questions, as well as constructive suggestions, which have certainly helped us to improve the quality of this manuscript.

Reviewers' comments:

Reviewer #1 (Remarks to the Author):

The comments raised by the reviewers have been well addressed. I recommend its publication on Nature Communications.

Reviewer #4 (Remarks to the Author):

Crystallographic review.

Recommend for publication. The new crystallographic data and information supplied by the authors has satisfactorily addressed all points raised in my original comments.

Reviewer #5 (Remarks to the Author):

The authors synthesized a series of Zr-based MOFs with ftw, scu, and lvt topology. Breakthrough experiments were carried out for two (ftw, scu) of the MOFs for three different alkane isomers (n-hexane, 2-methylpentane (2MP), 2,3-dimethylbutane (23DMB)). The breakthrough experimental results show that the compound 2 with scu topology can differentiate between mono-branched isomer from di-branched isomers. Ab initio molecular simulations were carried out to assess the origin of shape-selectivity, and the authors conclude that the separation mechanism is based on thermodynamic factors, which are corroborated by the adsorption isotherm, heats of adsorption, and in situ IR spectroscopy measurements.

Separation of hexane isomers is an important topic, and the design principles for the novel adsorbent material outlined in the experimental section of the manuscript is sound and justified. However, as far as the computational aspect of the paper is concerned, I do not think the paper is suitable for the publication in Nature Communications unless the following itemized questions are answered:

- 1) In breakthrough experiments, nC6, 3MP, and 23DMB were measured, but the energy barriers were calculated for nC6, 3MP, and 22DMB isomers. The authors need to provide calculation results for five hexane isomers. Actually, it would be the best for the authors to provide breakthrough experiments for all five hexane isomers and not only three.
- 2) The computational investigation only addresses the selectivity of compound 1 with ftw topology, which can differentiate n-hexane from mono-branched isomers. It would be good to include the results for compound 2 with scu topology which can differentiate mono-branched isomer from di-branched isomers.
- 3) Main Figure 4 (f): the data point for 23DMB above 70 is missing. The data point was clearly listed in the Supplementary Figure 36.
- 4) Supplementary Figure 4: how was the aperture size calculated? If using van der Waals representation, please list the size of the vdW radii for each atom.
- 5) Supplementary Table 9: please include all hexane isomers for T = 423 K.
- 6) From the reading of Supplementary Note 1, it appears that the simulations are carried out with VASP and performed periodic calculations. Are the models shown in Supplementary Figure 42 for periodic calculations? I find it strange to perform periodic calculations on a cluster model.
- 7) Supplementary Figure 42: In the caption it says "about to enter" please be exact. It would be good to provide geometries for all isomers investigated in this work.
- 8) Supplementary Figure 43: The window areas shown in d, e, f are based on ball-and-stick model, or van der Waals representation? If so please list what are the radii of the atoms.
- 9) Supplementary Figure 44: The authors need to look at more than two points on the potential energy surface (PES) that are formed by the ligands. It may be that the highest energy point of n-hexane was not sampled.
- 10) Supplementary Table 8: Why are the energy barriers for 2-methylpentane and 2,2-

dimethylbutane not available?

11) Page 59 in SI: What kind of classical calculations were carried out?

The main concern that I have with the computational work presented in this manuscript is the simulation results do not really support the experiments. The authors explicitly wrote in the manuscript that the separation mechanism is determined by thermodynamic principle, in which case the difference in the heats of adsorption drives the separation. However, the calculation presented in this work is, at least to me, applicable for the case if the size-exclusion principle is at work. In fact, it is very strange for the authors to perform the diffusion barrier calculations. In Figure 4(f), the heats of adsorption values show marked difference between linear, mono-, and di-branched isomers for compound 2, which suggests that the separation is thermodynamically-driven, and not by the size-exclusion principle.

Some minor points:

- Please change the units of all calculations from eV to kJ/mol if possible.
- Line 278, SI pg. 31: "N-hexane" ->"n-hexane"
- I suggest the authors to provide relevant computational input files to reproduce their work for others.

REVIEWERS' COMMENTS:

Reviewer #5 (Remarks to the Author):

The authors synthesized a series of Zr-based MOFs with ftw, scu, and lvt topology. Breakthrough experiments were carried out for two (ftw, scu) of the MOFs for three different alkane isomers (n-hexane, 2-methylpentane (2MP), 2,3-dimethylbutane (23DMB)). The breakthrough experimental results show that the compound 2 with scu topology can differentiate between mono-branched isomer from di-branched isomers. Ab initio molecular simulations were carried out to assess the origin of shape-selectivity, and the authors conclude that the separation mechanism is based on thermodynamic factors, which are corroborated by the adsorption isotherm, heats of adsorption, and in situ IR spectroscopy measurements. Separation of hexane isomers is an important topic, and the design principles for the novel adsorbent material outlined in the experimental section of the manuscript is sound and justified. However, as far as the computational aspect of the paper is concerned, I do not think the paper is suitable for the publication in Nature Communications unless the following itemized questions are answered:

Main Clarification:

We are grateful to the reviewer for the positive comments on the experimental section of this work, as well as valuable questions/suggestions. In **comment #12**, the reviewer points out his/her main concern:

“The main concern that I have with the computational work presented in this manuscript is the simulation results do not really support the experiments. The authors explicitly wrote in the manuscript that the separation mechanism is determined by thermodynamic principle, in which case the difference in the heats of adsorption drives the separation. However, the calculation presented in this work is, at least to me, applicable for the case if the size-exclusion principle is at work. In fact, it is very strange for the authors to perform the diffusion barrier calculations. In Figure 4(f), the heats of adsorption values show marked difference between linear, mono-, and di-branched isomers for compound 2, which suggests that the separation is thermodynamically-driven, and not by the size-exclusion principle.”

This comment indicates that there may have been a misunderstanding concerning an important aspect. In this work, we experimentally studied the separation of C₆ alkane isomers by two MOF compounds: compound **1** (Zr-bptc) and compound **2** (Zr-abtc). The reviewer is absolutely correct about the thermodynamically-driven separation process in the case of compound 2. As the reviewer mentioned, which we also believe, computational calculations of energy barriers would not provide further insight for such a process. This is why we did NOT calculate the diffusion barriers for compound **2** (it appears the reviewer thought the energy barrier calculations were performed on compound **2**, based on his/her comment #12). In contrast, our experimental results indicate that the separation by compound 1 is based on selective molecular exclusion (or size sieving). It adsorbs nHEX only and excludes all branched isomers. Diffusion plays an important role in this non thermodynamically-driven process and thus, energy barriers are important and are

calculated for compound **1**. We have revised the manuscript to emphasize the different mechanisms of the two MOFs.

1) In breakthrough experiments, nC6, 3MP, and 23DMB were measured, but the energy barriers were calculated for nC6, 3MP, and 22DMB isomers. The authors need to provide calculation results for five hexane isomers. Actually, it would be the best for the authors to provide breakthrough experiments for all five hexane isomers and not only three.

Response: The reviewer brings up a good point that the computationally calculated isomers for compound **1** were not an exact match to the experimentally tested isomers. Our initial computational models were simply selected to include isomers with different branching (namely linear, monobranched and dibranched species). The calculation results clearly show the trend: higher energy barrier with increasing branching of the isomers. However, to fully answer the reviewer's question, we have now performed additional computational calculations so that the same isomers are investigated by both experimental and computational methods. The calculated energy barriers for nHEX, 3MP, and 23DMB again support our experimental results that the separation in compound **1** is based on selective size exclusion and only nHEX can be adsorbed. Related contents in the manuscript have been updated (main text, Page 16-17 and Figure 5a). In addition, as suggested by the reviewer, we have also calculated and summarized the energy barriers for all five isomers. The related data are added to the Supplementary Information (Page 61-62, Supplementary Table 9 and Supplementary Figure 42-44). The results confirm that the energy barriers are too high for any branched isomers to diffuse into the pore.

With respect to experimental breakthrough measurements, we believe the use of ternary mixture as a feed is sufficient to reflect the separation capability of the materials. For compound **1**, our calculations indicate that any branched isomer would not diffuse into its pores, as a result of extremely high energy barrier. The ternary mixture selected for breakthrough experiments fully represents isomers of different degree of branching (linear, monobranched, dibranched), and the results confirm that only the linear isomer can be adsorbed. For compound **2**, our conclusion is that it shows thermodynamically-driven separation of C6 alkane isomers of different degrees of branching, which is important for improving RON of gasoline. Separation between two isomers with the same degree of branching (e.g. monobranched 2MP and 3MP or dibranched 23DMB and 22DMB) is out of the scope of this work, as 2MP and 3MP (or 22DMB and 23DMB) have very similar RON values. A previous work (Ref. #27) also confirmed that in such a case (separation via thermodynamic process), isomers with the same degree of branching (e.g. 2MP and 3MP, 22DMB and 23DMB) show very similar adsorption behavior.

2) The computational investigation only addresses the selectivity of compound **1** with ftw topology, which can differentiate n-hexane from mono-branched isomers. It would be good to include the results for compound **2** with scu topology which can differentiate mono-branched isomer from di-branched isomers.

Response: This comment relates to Comment #12 and our "Main Clarification". Here we would like to state again that computational calculations are only needed and done for compound **1** (Zr-bptc), as the separation by compound **2** was experimentally proven to be thermodynamically driven and calculation of energy barriers would not provide any further insight.

3) Main Figure 4 (f): the data point for 23DMB above 70 is missing. The data point was clearly listed in the Supplementary Figure 36.

Response: Figure 4(f) has been corrected and updated.

4) Supplementary Figure 4: how was the aperture size calculated? If using van der Waals representation, please list the size of the vdW radii for each atom.

Response: The aperture size was calculated from the shortest distance connecting the opposite sides of the pore window which exclude van der Waals radii (vdW radii used: C: 1.7 Å, H: 1.20 Å, O: 1.52 Å). This information has been added to Supplementary Figure 4.

5) Supplementary Table 9: please include all hexane isomers for T = 423 K.

Response: Energy barriers for all five isomers in compound **1** have been calculated and Supplementary Table 9 has been updated accordingly.

6) From the reading of Supplementary Note 1, it appears that the simulations are carried out with VASP and performed periodic calculations. Are the models shown in Supplementary Figure 42 for periodic calculations? I find it strange to perform periodic calculations on a cluster model.

Response: We agree that it seems strange to use a periodic calculation for a cluster model. But, it is nonetheless a well-accepted standard textbook approach, see e.g. *Atomistic Computer Simulations* by V. Brazdova and D.R. Bolwer (Wiley-VCH). There are in fact several good reasons for using this approach. The main one for many authors in the community has to do with the basis set completeness. Periodic codes almost exclusively use plane-waves as basis set and a *single* adjustable parameter allows one to precisely control the size and completeness of the basis, which is a huge advantage over any localized basis set. Another good reason is that localized basis sets struggle from Pulay corrections when calculating forces and pressure as well as basis set superposition errors—problems that plane-wave codes completely avoid. However, besides these common reasons, we had one more reason that was important to us. Small molecules interact with the MOF framework often via van der Waals forces and to accurately capture those important interactions specialized exchange-correlation functionals are required. During the initial investigation of our work on MOFs we tested the exchange-correlation functional vdW-DF, but this functional is not readily available in localize-basis set codes. Although in the end we used the standard PBE functional (since van der Waals effects were negligible on our barriers, which are to a large extent determined by steric hindrances), it was nonetheless crucial that we did some initial testing with a van der Waals-including functional. For all the reasons mentioned above, we thus chose the plane-wave code VASP for this project.

Note that we did make our unit cell large enough such that period replica of the cutout clusters are separated by at least 10 Å to minimize spurious interactions (a commonly accepted value in the community is typically 8 Å). We have included a sentence in Supplementary Note 1 to specify the amount of separation we have used.

7) Supplementary Figure 42: In the caption it says “about to enter” please be exact. It would be good to provide geometries for all isomers investigated in this work.

Response: We have added all relevant images of studied isomers to Supplementary Figure 42 and the caption wording in question has been changed from “about to enter” to “at the entrance of” for better clarity. Additionally, we have added more details regarding how the geometries were chosen to Supplementary Note 1 (Supplementary Information, Page 61).

8) Supplementary Figure 43: The window areas shown in d, e, f are based on ball-and-stick model, or van der Waals representation? If so please list what are the radii of the atoms.

Response: They are based on ball and stick models for simplification in order to easily view the differences in pore size.

9) Supplementary Figure 44: The authors need to look at more than two points on the potential energy surface (PES) that are formed by the ligands. It may be that the highest energy point of n-hexane was not sampled.

Response: The reviewer makes a very good point and ideally more than two points on the PES should be tested. However, we were able to estimate the position of the point of highest barrier through prior ground-state (0 K) transition-state searches. Our simulations indicated the transition states to be those depicted in Supplementary Figure 42 where the isomer experiences the highest barrier. We argue that the point of highest barrier does not drastically change when going from 0 K to the experimental temperature used in our AIMD simulations, which is a reasonable assumption. Thus, the tactical decision was made to estimate the transition state via a transition-state search algorithm and proceed with the AIMD calculations (described in Supplementary Note 1) for the PESs shown in Supplementary Figure 42, which correspond to the starting image and the transition-state image. We have included more details in Supplementary Note 1 to address this reviewer’s comment and further clarify how we chose the PESs studied.

10) Supplementary Table 8: Why are the energy barriers for 2-methylpentane and 2,2-dimethylbutane not available?

Response: Supplementary Table 8 has been updated to more relevantly address a main point of the calculations, the necessity for AIMD calculations. Ground-state (0 K) temperature barriers were not calculated for 2MP and 2,2DMB because nHEX still had too high of a barrier to explain equilibrium diffusion through the pore with the three 0 K models studied. We now use Supplementary Table 8 to explain the need for the AIMD temperature-incorporated energy barrier calculations. Energy barriers for all branched isomers can be found in Supplementary Table 9 and Supplementary Fig. 44. Additionally, Supplementary Table 8 has been changed to kJ/mol units per suggestion of the reviewer.

11) Page 59 in SI: What kind of classical calculations were carried out?

Response: Due to the large unit cell size of compound **1** (~850 atoms), classical force-field calculations were carried out originally. Using the GULP software [J.D. Gale, JCS Faraday Trans., 93, 629 (1997)] with the UFF [Rappe, A. K., Casewit, C. J., Colwell, K. S., Goddard, W. A. & Skiff, W. M. UFF, J. Am. Chem. Soc. 114, 10024 (1992)] and UFF4MOF [Addicoat, M. A., Vankova, N., Akter, I. F. & Heine, T, J. Chem. Theory Comput. 10, 880 (2014)] force fields, we calculated the energy barrier of nHEX entering the pore of compound **1** using the built-in GULP transition-state methods and in addition by manually calculating the trajectory to enter the pore. Both methods resulted in an unreasonably high energy barrier for nHEX to go through to MOF pore window of approximately 500 kJ/mol—this number is too high, as we know from experiment that nHEX easily traverses the MOF pore window at room temperature. This led us to implement the more accurate *ab initio* methods defined in Supplementary Note 1 and our results *a posteriori* validate this decision.

We have thoroughly updated Supplementary Note 1 to provide more details about all the calculations we have performed. We have also streamlined Supplementary Note 1 to provide information that is truly supplementary to the main manuscript, rather than repeating it. The classical simulations mentioned earlier were originally meant to document how we approached the problem of finding suitable barriers. We started with this low-accuracy method to gain initial insight. In the end, however, we did find that higher accuracy *ab initio* calculations were necessary and we switched our entire approach over to *ab initio* modeling. As such, none of the initial classical results are of any relevance. In fact, the reviewer's remark led us to conclude that the mentioning of this initial method is distracting from main story of our simulations. In addition, they do not aid any understanding. As such, we decided in this updated resubmission to remove any reference to classical simulations during the initial phase of our investigation.

12) The main concern that I have with the computational work presented in this manuscript is the simulation results do not really support the experiments. The authors explicitly wrote in the manuscript that the separation mechanism is determined by thermodynamic principle, in which case the difference in the heats of adsorption drives the separation. However, the calculation presented in this work is, at least to me, applicable for the case if the size-exclusion principle is at work. In fact, it is very strange for the authors to perform the diffusion barrier calculations. In Figure 4(f), the heats of adsorption values show marked difference between linear, mono-, and di-branched isomers for compound **2**, which suggests that the separation is thermodynamically-driven, and not by the size-exclusion principle.

Response: Please see “Main Clarification” provided at the beginning of this response letter. We are particularly thankful to the reviewer for this comment, and hope that our responses and revisions have clarified the misunderstanding and addressed the question.

Some minor points:

- Please change the units of all calculations from eV to kJ/mol if possible.

Response: All barriers and mention of barriers in the SI and MS have been converted from eV to kJ/mol.

- Line 278, SI pg. 31: “N-hexane” ->“n-hexane”

Response: This has been corrected.

- I suggest the authors to provide relevant computational input files to reproduce their work for others.

Response: Related files for the AIMD calculations have been provided.

REVIEWERS' COMMENTS:

Reviewer #5 (Remarks to the Author):

Authors have answered all the questions by updating the main manuscript and the SI, so I recommend the publication of the manuscript in Nature Communications given that the authors address the following comments:

Minor comments:

1) Authors may wish to cite some of the recent computational and/or experimental papers related to size-exclusion and thermodynamically-driven C6 isomer separation using MOFs.

2) Please change all "N-hexane" to "n-hexane" where n is italicized.

3) pg. 4, "none of them have reached the performance level of the benchmark material, zeolite 5A, with respect to both adsorption capacity and selectivity under similar industrial relevant conditions."  this is clearly not true, as Fe₂BDP synthesized by Long group has shown to have both higher selectivity and capacity than Zeolite 5A. Please revise the sentence.

RESPONSE TO REVIEWER'S COMMENTS

Reviewer #5 (Remarks to the Author):

Authors have answered all the questions by updating the main manuscript and the SI, so I recommend the publication of the manuscript in Nature Communications given that the authors address the following comments:

Minor comments:

1) Authors may wish to cite some of the recent computational and/or experimental papers related to size-exclusion and thermodynamically-driven C6 isomer separation using MOFs.

Response: More recent related references have been cited (refs. 30-33).

2) Please change all "N-hexane" to "n-hexane" where n is italicized.

Response: Corrections have been made throughout the manuscript.

3) pg. 4, "none of them have reached the performance level of the benchmark material, zeolite 5A, with respect to both adsorption capacity and selectivity under similar industrial relevant conditions."  this is clearly not true, as Fe2BDP synthesized by Long group has shown to have both higher selectivity and capacity than Zeolite 5A. Please revise the sentence.

Response: We have revised the text accordingly (main text, pg.4 paragraph 2).